# Pathogen infection induces sickness behaviors through neuromodulators linked to stress and satiety in *C. elegans*

Sreeparna Pradhan[1,2], Gurrein K. Madan[1,2], Di Kang[1], Eric Bueno [1],
Adam A. Atanas[1], Talya S. Kramer[1], Ugur Dag[1], Jessica D. Lage[1],
Matthew A. Gomes[1], Alicia Kun-Yang Lu[1], Jungyeon Park[1] & Steven W. Flavell [1] ✉

When animals are infected by a pathogen, peripheral sensors of infection signal to the brain to induce adaptive behavioral changes known as sickness behaviors. While the pathways that signal from the periphery to the brain have been intensively studied, how central circuits are reconfigured to elicit these behavioral changes is not well understood. Here we find that neuromodulatory systems linked to stress and satiety are recruited during chronic pathogen infection to alter the behavior of *Caenorhabditis elegans*. Upon infection by the bacterium *Pseudomonas aeruginosa* PA14, *C. elegans* decrease feeding, then display reversible bouts of quiescence, and eventually die. The ALA neuron and its neuropeptides FLP-7, FLP-24, and NLP-8, which control stress-induced sleep in uninfected animals, promote the PA14-induced feeding reduction. However, the ALA neuropeptide FLP-13 instead delays quiescence and death in infected animals. Cell-specific genetic perturbations show that the neurons that release FLP-13 to delay quiescence in infected animals are distinct from ALA. A brain-wide imaging screen reveals that infection-induced quiescence involves ASI and DAF-7/TGF-beta, which control satiety-induced quiescence in uninfected animals. Our results suggest that a common set of neuromodulators are recruited across different physiological states, acting from distinct neural sources and in distinct combinations to drive state-dependent behaviors.

The physiology of an animal profoundly influences its behavior[1,2]. For example, when animals are hungry, sick, or injured, they alter their feeding and exploratory behaviors to maximize their chances of survival. The interoceptive pathways that allow physiological signals in the body to influence behavior have been the subject of intense investigation in recent years, revealing several routes of signaling from the body to the brain[3,4]. However, how neural circuits in the brain are modulated by these physiological changes to alter behavior remains poorly understood.

Infection by a pathogen changes many aspects of an animal's behavior, collectively referred to as "sickness" behaviors, that help animals restore bodily homeostasis and adapt their behavior during illness. These commonly include reduced appetite, fever, and lethargy[5-7]. In mammals, multiple pathways relay information about infection to the central nervous system (CNS) to induce these infection-induced behavioral changes. Peripheral nerve endings in distal tissues, for example those of the vagus nerve, receive molecular signals pertaining to infection and propagate this information to the CNS[3]. In addition, cytokines produced by circulating immune cells can also act on CNS neurons and cells that line the blood-brain barrier to signal infection[8]. CNS brain regions that are modulated by infection and, in turn, alter behavior include the nucleus of the solitary tract[9], the

[1]Howard Hughes Medical Institute, Picower Institute for Learning & Memory, Department of Brain & Cognitive Sciences, Massachusetts Institute of Technology, Cambridge, MA, USA. [2]These authors contributed equally: Sreeparna Pradhan, Gurrein K. Madan. ✉e-mail: flavell@mit.edu

insular cortex[10,11], and subsets of hypothalamic nuclei[12]. Neurons in these brain regions can control aspects of motivated behavior, as well as autonomic functions like the control of body temperature[9,10,12,13]. How these neurons are embedded in a broader circuit architecture that is modulated by infection to control behavior remains an area of active investigation.

The nematode *C. elegans* provides a system where it should be possible to connect physiological changes, like infection, to precise changes in neural circuit function that impact behavior. The *C. elegans* nervous system consists of 302 uniquely identifiable neurons connected through a fully defined wiring diagram[14–16]. These neurons control a well-defined set of motor outputs, including locomotion, feeding, and egg-laying, as well as more complex behaviors like foraging and mating. *C. elegans* modulates its behavior based on its physiological state[17–19]. For example, acute noxious stressors, like heat shock or UV exposure, lead to bouts of sleep-like quiescence, a behavioral state that depends on the ALA neuron releasing several somnogenic neuropeptides[20–22]. Changes in food availability, ingestion, and metabolism impact the animal's foraging behaviors[23–26], which depend on neuromodulators like serotonin, octopamine, and insulin-like peptides. In addition, re-feeding after food deprivation induces satiety quiescence, which requires the release of DAF-7/TGF-beta from the ASI neuron[27,28]. Thus, the neuromodulatory systems of *C. elegans* allow the animal to adapt its behavior to its ongoing physiological state.

*C. elegans* physiology changes as animals become infected by pathogens. When they ingest pathogenic bacteria, these pathogens can populate the *C. elegans* gut, damage tissues, and cause death[19]. The opportunistic human pathogen *Pseudomonas aeruginosa* strain PA14 has been especially well-studied in this regard. PA14 can release toxins that cause acute injury[29]. In addition, it populates the intestine, where it causes dysbiosis and ruptures the intestinal lining to kill animals[29]. *C. elegans* has a multi-level defense system to fight infection. Innate immune pathways activated by PA14 infection trigger activation of adaptive stress responses and the production of anti-microbial peptides, which help maintain bodily homeostasis[30,31]. Furthermore, *C. elegans* exhibits PA14-induced behavioral changes that defend against infection[19]. Infected animals display an avoidance behavior where they leave PA14 bacterial lawns after hours of exposure. This lawn avoidance is triggered by the production of the DAF-7/TGF-beta humoral signal from ASJ neurons[32,33] and FLP-1 neuropeptides from AVK neurons[34]. PA14-exposed animals form associative memories where they learn to avoid the PA14 olfactory cues that were present during infection, so that they navigate away from PA14 in the future[35,36]. This olfactory learning involves serotonin and insulin-like peptides that act on sensory processing neurons to modulate olfactory navigation[35,37]. These studies suggest that neuromodulators play an important role in controlling PA14-induced behavioral changes.

Here, we show that *C. elegans* animals chronically infected with PA14 exhibit reduced feeding, followed by quiescence, and then death. The PA14-induced feeding reduction is mediated by the ALA neuron, which controls stress-induced sleep in uninfected animals. In agreement with ALA's role in suppressing feeding and locomotion in stressed animals, we found that several neuropeptides that are released by ALA suppress feeding in PA14-infected animals. However, one of these ALA-released neuropeptides, FLP-13, that contributes to stress-induced sleep, is instead released by other neurons in infected animals. In contrast to its previously reported somnogenic role, FLP-13 acts to delay PA14-induced quiescence and death. PA14-induced quiescence requires ASI and DAF-7/TGF-beta, a pathway known to control satiety quiescence in uninfected animals. Taken together, these results suggest that stress- and satiety-associated neuromodulatory systems contribute critically to PA14-induced behavioral changes. Our results are consistent with a model in which individual neuromodulators do not map onto individual internal states but are

deployed from different sources and in different combinations to control distinct internal states, like stress, satiety, and infection.

## Results

### Chronic infection by live PA14 bacteria causes reduced feeding and bouts of quiescence

To determine the time course of PA14 infection-induced behavioral changes and death, we first performed a survival assay to characterize the onset of death after animals were transferred to PA14. Animals started dying after two days of first being exposed to PA14, with more than 90% of animals dead by the fifth day (Fig. 1A). Next, we quantified behavioral changes leading up to this point. PA14 infection is known to impact bacterial lawn leaving and olfactory learning[19], but the impact on other motor programs was unknown. To identify early behavioral changes, we characterized how PA14 infection alters the different motor outputs of the animal over the first day of infection (Fig. 1B, Fig S1A-C). Animals were infected on large lawns of bacteria where they spend most of their time on the lawn, but have the choice of leaving. Given the direct relevance of feeding to pathogen ingestion, we first quantified feeding (or pharyngeal pumping) behavior (Fig. 1B). Over the first 20 h of infection, animals exhibited a progressive decrease in feeding not observed in control animals exposed to non-pathogenic *E. coli* OP50 bacteria (Fig. 1B). A previous study noted this decline in feeding rates in PA14-infected animals after 24–48 h of infection[29], but the basis of this behavioral change was not further characterized.

We also quantified other motor programs (Fig S1A–C) and found a slight decrease in speed and egg-laying[38] as infection progressed, but no changes in defecation. We tested whether the reduction in egg-laying could indirectly cause the reduction in feeding by examining if animals that have been sterilized by FUDR treatment have altered feeding rates. However, FUDR-treated animals had no change in pharyngeal pumping rates (Fig S1D). Interestingly, we observed an increase in egg-laying upon initial exposure to PA14 (Fig S1B). To test whether this was a response to a live infection, we examined egg-laying during the first hour of exposure to UV-killed PA14. This did not induce the increase in egg-laying observed in response to infectious PA14, suggesting that this behavioral changes requires live infectious bacteria (Fig S1E).

Notably, starting at two days of infection, we observed that animals frequently exhibited bouts of quiescence (defined as no feeding and no movement for >20 s) (Fig. 1A, Fig S1F–G show OP50 controls). Taken together, these results indicate that PA14 infection suppresses the animal's feeding behavior over the first day of infection, which is followed by quiescence and eventually death.

We next focused on the PA14-induced decrease in feeding because of its signature as an early behavioral change after PA14 exposure and its possibly relevant role in combating infection. We used multiple approaches to test whether the PA14-induced feeding reduction required an active infection. First, we compared the feeding behavior of animals exposed to live PA14 to animals exposed to UV-killed PA14 for 20 h (Fig. 1C). Animals fed on live PA14 as well as UV-killed PA14 were both transferred to live PA14 plates before assaying feeding rates to control for the acute effects of exposure to PA14 odors or metabolites. Animals previously exposed to live PA14 displayed a reduction in feeding, whereas animals exposed to UV-killed PA14 displayed high feeding rates matching OP50 controls (Fig. 1C). This suggests that only live PA14, capable of infecting *C. elegans*, elicit a decrease in feeding. We also examined the behavioral responses of mutant animals lacking key regulators of innate immunity (*pmk-1* and *tir-1*), which have more severe infections[39,40] (Fig. 1D). Both of these mutants displayed exaggerated feeding reductions after PA14 exposure compared to wild-type, providing evidence that innate immunity reduces the impact of PA14 infection on feeding behavior. This finding aligns with the established function of innate immunity pathways in defending against PA14 infection in general[30,39]. Finally, we examined

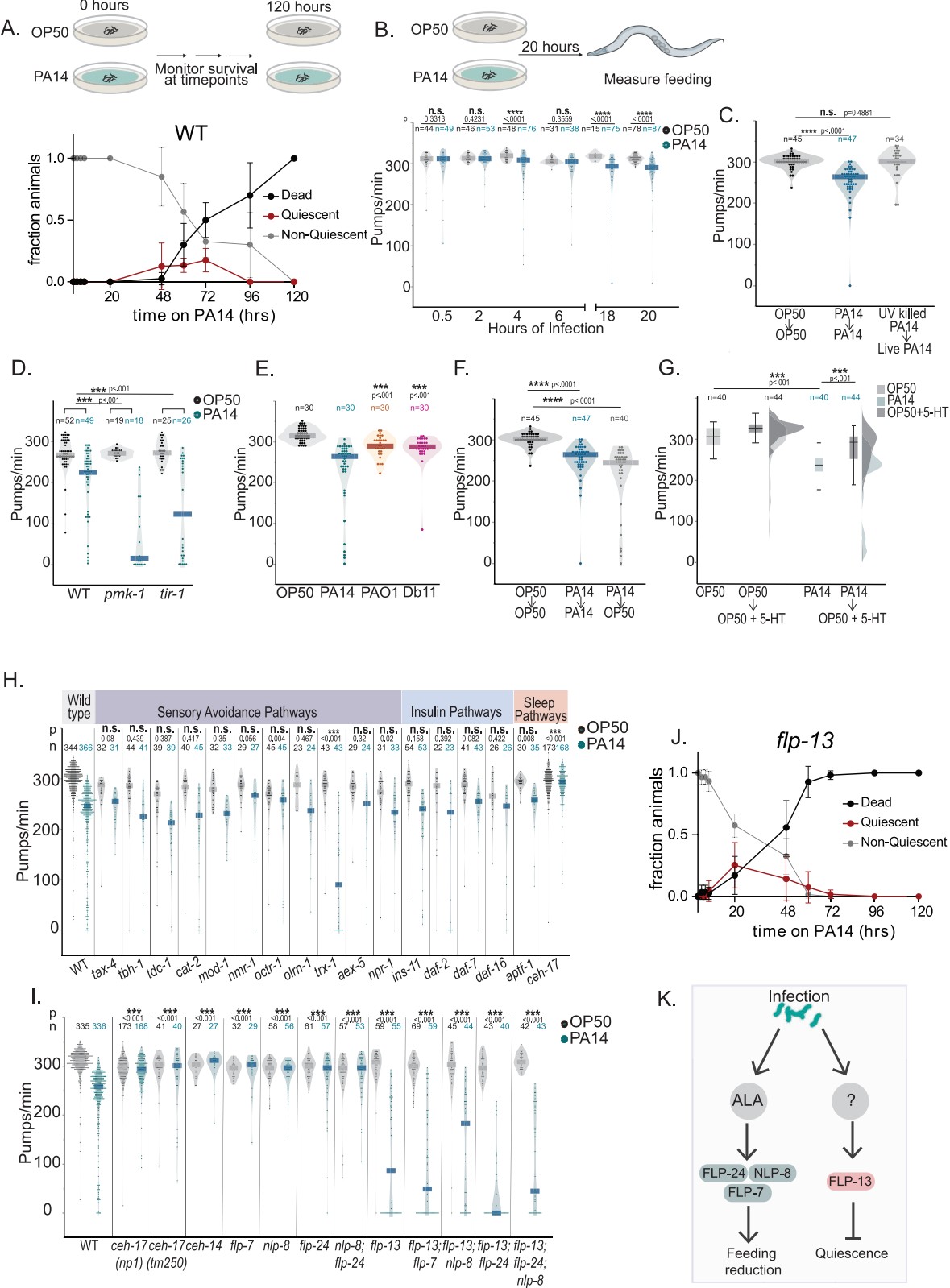

the behavioral effects of less infectious variants of *Pseudomonas* and other milder pathogens. Exposure to *Pseudomonas* PAO1, which is non-infectious under our culture conditions, and *Serratia* Db11, a bacteria with reduced pathogenicity[29], did not induce feeding decreases to the same extent as PA14 (Fig. 1E). Taken together, these results indicate that a live PA14 infection over 20 h causes a reduction in feeding in *C. elegans*.

We further characterized this PA14-induced decrease in feeding. First, we tested whether the PA14-induced decrease in feeding is specific to PA14 bacteria or, alternatively, a more general reduction in feeding. To do so, we examined whether transferring infected animals to *E. coli* OP50 could recover feeding rates. We found that PA14-infected animals still displayed reduced feeding on OP50, suggesting that a persisting PA14 infection could still reduce feeding rates on

**Fig. 1 | PA14 infection induces behavioral changes, with a reduction in feeding as an early hallmark. A** Time course of WT behavior and viability upon infection. Top: Experimental design. Bottom: data. Error bars indicate standard deviation. For each genotype, n = 10 animals per replicate, 6 replicates. **B** Top: experimental design. Bottom: feeding rates (pumps/min) of WT animals on indicated bacteria. Indicated statistical comparison were Wilcoxon test. **C** Feeding of animals on indicated bacteria for 20 h, and then assayed on the indicated bacteria 1 h after transfer. PA14 killing was by UV light. p values are with Wilcoxon rank sum test. **D** Feeding rates of indicated genotypes. **E** Feeding rates of WT animals on indicated bacteria. **F** Feeding rates of animals exposed to the indicated bacteria for 20 h and then assayed on the indicated bacteria 1 h after transfer. p values are with Wilcoxon rank sum test. Same OP50 to OP50 and PA14 to PA14 controls were used in 1 C and 1 F. **G** Feeding in response to exogenous serotonin (15 mM) for 10 min. Raincloud distributions are on right. *p* values are with Wilcoxon rank sum test. Box indicates interquartile range; whiskers show 5th and 95th percentiles; line shows median. Effects of 20 h of PA14 infection on feeding for mutant strains. The same *ceh-17* animals are plotted in (**H, I**). **J** Behavior and viability upon infection, shown as (**A**). Error bars indicate standard deviation. For each genotype, n = 10 animals per replicate, 6 replicates. Median quiescence for *flp-13* significantly different vs WT animals in (**A**). Two-tailed Mann–Whitney, *p < 0.05. Median time point at which 50% animals died not significantly different for *flp-13* vs WT animals in Fig. 1A. Two-tailed Mann–Whitney, n.s. (p = 0.057). **K** Summary of neuropeptide function. For (**B**–**F**), (**H**–**I**), n's are individual animals and are indicated on figure. Dots are individual animals; dark bar indicates mean. n.s. indicates p > 0.05. For (**D, E**), (**H, I**), asterisks above each condition indicate whether the feeding decrease was significantly different from the decrease in day matched WT controls. Empirical p-values indicated on figure. Bonferroni correction was applied. Source data are provided as a source data file.

other bacterial food sources (Fig. 1F). We also tested whether the reduction in feeding could be due to damage to pharyngeal tissues making animals unable to pump at high rates. We exposed PA14-infected animals to exogenous serotonin, which is known to elevate pumping rates[41]. Exogenous serotonin application caused an increase in pumping in PA14-infected animals, suggesting that these animals are able to pump at higher rates, but do not do so under PA14-infected conditions (Fig. 1G). To extend our findings, we used video recordings and counted pumps at a slowed down speed during video playback. This quantification method also revealed the reduction in feeding upon PA14 infection (Fig S1H). Interestingly, this further showed that the intervals between successive grinder movements were longer in infected animals due to an increased duration of time between the ends of some pharyngeal pumps and the initiation of subsequent pumps (Fig S1I, S1J, recordings at 60fps; higher-resolution measurements will be necessary to capture millisecond-resolution changes in pumping in infected animals; Fig. S1K shows fluorescent bacteria ingestion).

We next asked whether this decrease in feeding was related to the robust PA14 bacterial accumulation and bloating of the gut that has been reported in PA14-infected animals[42]. To test this, we examined the feeding rates of three mutants (*pbo-1*, *aex-2*, and *flr-1*) known to have deficits in the defecation program, leading to bloating of the gut even during OP50 consumption[42,43]. Indeed, we observed that all three mutants pumped at a lower rate than wild-type animals on OP50 (Fig S1L), raising the possibility that intestinal bloating could play a role in reducing feeding behavior in PA14-infected animals.

To identify neural mechanisms that underlie these effects, we examined how PA14 impacts feeding behavior in a panel of 17 mutant strains lacking signaling pathways known to be involved in stress- or pathogen-related behavioral changes (Fig. 1H). While not exhaustive, this list included mutants lacking key neuromodulators (*tbh-1*, *tdc-1*, etc.), neuromodulatory receptors (*octr-1*), components of sensory signaling pathways (*tax-4*, etc.), and more. Several of these pathways have been linked to either PA14 responses or feeding regulation[19,32–37]. For each mutant, we asked whether PA14 infection reduced their pumping rates compared to baseline feeding rates on OP50. Of all the mutants tested, only *ceh-17* mutants, which lack functional ALA and SIA neurons, failed to reduce their feeding after PA14 infection. Animals lacking *trx-1*, which encodes a Thioredoxin important for PA14 avoidance behaviors[44], showed an exaggerated decrease in feeding. *ceh-17* animals exhibited normal baseline feeding rates on OP50 bacteria and normal feeding upon initial exposure to PA14 (Fig. S2A). A second null allele of *ceh-17* showed the same phenotype (Fig. 1I). Taken together, these data suggest that that the PA14-induced feeding decrease does not require many signaling pathways previously implicated in stress and pathogen responses, but it does require the *ceh-17* gene.

*ceh-17* encodes a homeobox transcription factor that is required for the specification of ALA and SIA neurons[45,46]. Due to the misspecification of ALA, these animals are known to be defective in stress-induced sleep[22,46,47], but their responses to PA14 infection had not been previously investigated. To test whether the *ceh-17* phenotype was due to ALA misspecification, we examined mutant animals lacking *ceh-14*, which have a defect in ALA specification but normal SIA development[46]. Like *ceh-17* mutants, *ceh-14* mutants did not show the PA14-induced suppression of feeding, suggesting that ALA is likely critical for the PA14-induced feeding reduction (Fig. 1I). *aptf-1* mutants lacking another stress-induced sleep neuron, RIS[48,49], displayed normal feeding decreases upon infection (Fig. 1H). We examined whether *ceh-17* animals were defective in other PA14-induced behavioral changes by quantifying PA14-induced lawn leaving in these mutants. *ceh-17* mutants displayed normal PA14 lawn leaving, matching wild-type controls (Fig S2B). This suggests that ALA is not required for PA14 lawn leaving and that the deficit in the feeding response is not due to a difference in PA14 lawn exposure. Together, these results suggest that ALA is specifically required for the PA14-induced feeding reduction.

Stress-induced sleep, which is controlled by ALA, shares a similarity with PA14 infection insofar as feeding rates are reduced during both states, albeit more robustly during stress-induced sleep. Given this parallel, we next tested whether the ALA-expressed neuropeptides that promote stress-induced sleep were required for the PA14-induced feeding decrease (Fig. 1I). Three ALA-expressed neuropeptides have been previously shown to have overlapping roles in suppressing motor outputs during sleep: *flp-13*, *flp-24*, and *nlp-8*[20,22]. Mutants lacking *flp-24* or *nlp-8* did not display reduced feeding upon PA14 infection, suggesting that each of these neuropeptides is required for the PA14-induced feeding decrease. In addition, the ALA neuropeptide *flp-7*, previously not reported to have a role in sleep, was also required for the infection-induced feeding decrease (Fig. 1I).

In contrast, animals lacking the neuropeptide gene *flp-13* displayed an exaggerated reduction in feeding upon PA14 infection. In addition, a large proportion of the *flp-13* animals displayed reversible quiescence bouts after 20 h of infection (Fig. 1J, Fig S1F–G, Supp Videos 1-2 show examples, under recording conditions used for pumping quantification), similar to the quiescence phenotype that was observed in wild-type animals much later in the course of infection (Fig. 1A). We performed survival assays to characterize the onset of quiescence and death in *flp-13* animals and found that animals displayed peak quiescence after 20 h of infection, and more than 90% animals died by 72 h, which is greatly accelerated compared to WT animals (Fig. 1J, A). We also characterized survival and onset of quiescence in *ceh-17* animals (Fig S2C) and found that *ceh-17* animals displayed quiescence and death at time points similar to wild-type animal, though the fraction of animals that were quiescent was reduced. Together, these results suggest that the ALA neuron and several neuropeptides are required for the PA14-induced pumping decrease, while the loss of *flp-13* accelerates the onset of quiescence and death in PA14-infected animals (Fig. 1K).

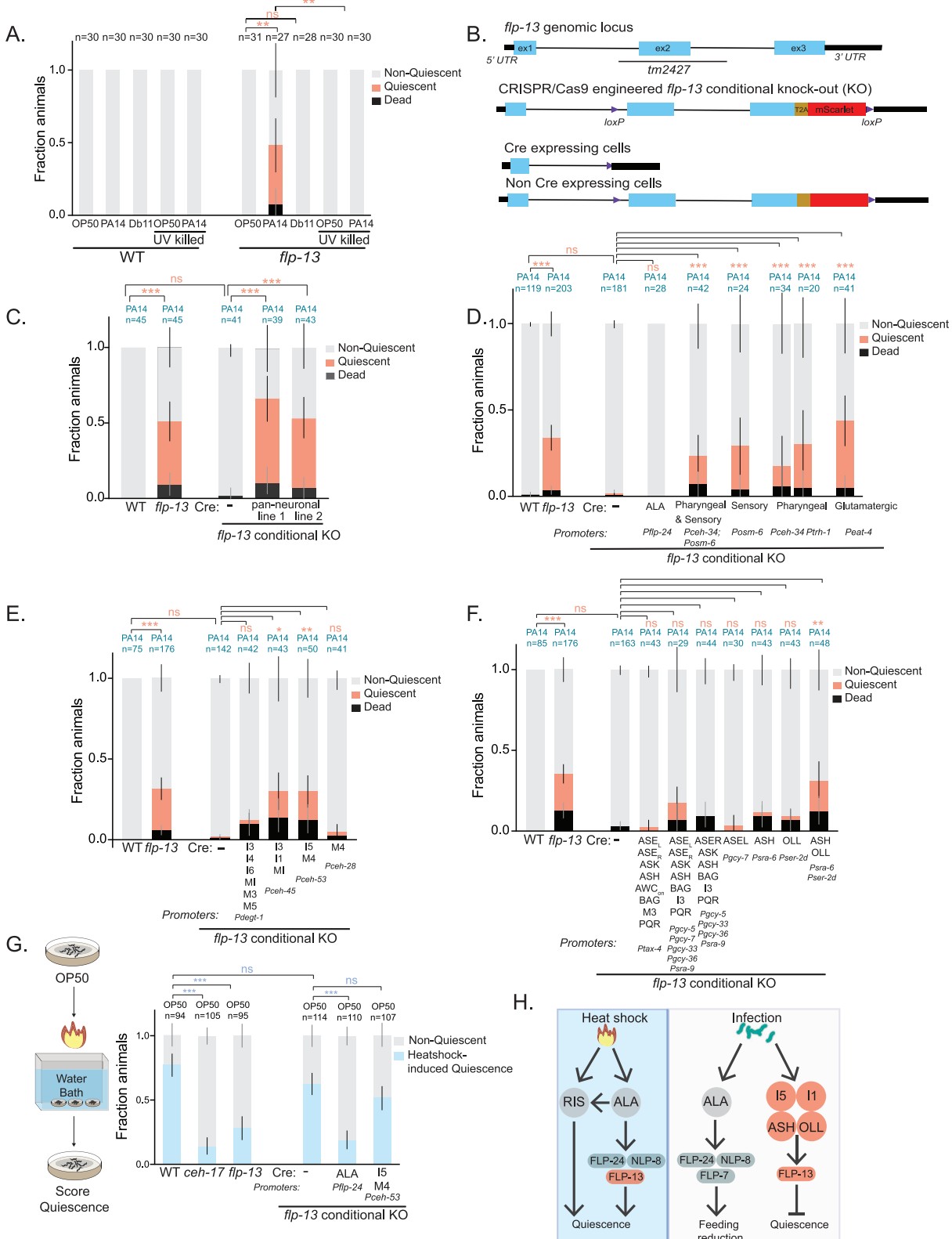

## FLP-13 release from non-pharyngeal sensory neurons and pharyngeal neurons prevents PA14-induced quiescence and death

We next examined if this accelerated quiescence was dependent on infection by live bacteria. PA14-induced quiescence in *flp-13* mutants was not seen on UV-killed PA14 (Fig. 2A), suggesting that this phenotype reflects a response to a live infection. Additionally, infection with a less virulent pathogen, *Serratia* Db11, did not lead to accelerated

quiescence like that observed in *flp-13* mutants on PA14 (Fig. 2A). Thus, the early onset of quiescence in *flp-13* mutants is a specific consequence of live PA14 infection. We next tested whether specific virulence factors expressed by PA14 were essential to drive this quiescence and indeed found that loss of *Pseudomonas* transcription factors *lasR* and *rhIR*, as well as the quorum sensing signaling molecule *pqsR*[50–52], led to a suppression of quiescence in *flp-13* animals (Fig S3A;

**Fig. 2 | *flp-13* functions in sensory and pharyngeal neurons to prevent PA14-induced quiescence and death. A** PA14-induced quiescence and death in indicated genotypes, shown as fraction of assayed animals. Animals were transferred to indicated bacteria as Day 1 adults. PA14 killing was by UV light. **B** Cartoon depicting CRISPR/Cas9-engineered *flp-13* conditional knockout allele. In this allele (*syb6180 syb6395*), loxP sites were positioned in the indicated locations and a t2a-mScarlet fluorescent reporter cassette was also inserted. Blue boxes indicate exons, black lines indicate introns. **C** PA14-induced quiescence and death in animals of the indicated genotypes after 20 h of infection, shown as fraction of assayed animals. The pan-neuronal promoter was *Ptag-168*. **D** PA14-induced quiescence and death in indicated genotypes after 20 h of infection, displayed as in (**C**). Here, a series of broadly-expressed promoters were used to drive expression of Cre recombinase. **E, F** PA14-induced quiescence and death in indicated genotypes after 20 h of infection, displayed as in (**C**). The neurons listed are those with intersecting expression of *flp-13* and the cell-specific promoter driving Cre expression. **G** Stress-induced sleep assays for the indicated genotypes. Data are shown as fraction of quiescent animals 30 min after a 35 min heat shock at 30 C. Note that the genotypes used here are identical to a subset of those used in (**E, F**). **H** Context-dependent functions of FLP-13, comparing heat shock versus infection. For (**A**), (**C**), (**D**), (**E**), (**F**), and (**G**), error bars represent the 95% confidence interval for the bootstrapped mean fraction values of animals that were quiescent, non-quiescent, and dead, respectively. Shown data was collected across multiple days but statistical comparisons were computed between day-matched experimental groups. **p < 0.01,***p < 0.001, fraction quiescence compared via Chi-square test with Bonferroni correction. Source data are provided as a source data file.

survival on OP50 shown in Fig. S3B). These results indicate that the quiescence observed in infected *flp-13* animals depends on a live infection of PA14 with intact virulence pathways.

One possible explanation for the exaggerated PA14-induced behavioral responses in *flp-13* mutants could be that the *flp-13* mutants fail to properly grind bacteria, leading to greater accumulation of live bacteria in the intestine[39,53]. To test this, we examined whether *flp-13* animals were defective at grinding and killing GFP-tagged OP50 during food ingestion, a well-established assay (Fig S3C). While two known grinder-defective mutants (*tnt-3* and *phm-2*) displayed dramatic increases in accumulation of live OP50-GFP in the intestine, *flp-13* animals did not display any deficit relative to wild-type animals (Fig S3C). This suggests that *flp-13* mutants grind bacterial food normally. To further characterize the *flp-13* mutant phenotype, we also examined whether other PA14 behavioral responses such as lawn leaving depended on *flp-13*. However, *flp-13* mutants displayed normal PA14 lawn leaving rates (Fig S3D), suggesting that the acceleration of quiescence and death in *flp-13* animals is not due to a difference in PA14 lawn exposure. Another possible explanation could be that *flp-13* animals have a reduced lifespan on all food sources. To test this, we performed lifespan assays on WT and *flp-13* mutants exposed to non-pathogenic OP50 bacteria (Fig S3B). In this assay, *flp-13* animals actually had a slightly extended lifespan compared to wild-type animals. A recent study also reported similar results where they found that FLP-13 from GABAergic neurons regulates lifespan extension[54].

We next determined the functionally relevant sites of FLP-13 release that impact the PA14 response. To test this, we used CRISPR/Cas9 genome editing to generate a conditional knockout allele of *flp-13* in which loxP sites were placed before the second exon and after the last exon of the endogenous gene (Fig. 2B). Expression of Cre Recombinase should delete the majority of the gene's coding sequence, attenuating *flp-13* function in cells with Cre expression. In addition, a t2a-mScarlet fluorescent reporter was inserted before the native stop codon of *flp-13* to visualize the gene's expression pattern (Fig. S3E). In the absence of Cre recombinase, this strain displayed a normal PA14-induced feeding decrease, matching wild-type controls (Fig. 2C). Pan-neuronal Cre expression in this strain caused the same phenotype observed in *flp-13(tm2427)* null animals where PA14 infection induced quiescence over an accelerated 20-h timeframe (Fig. 2C; note that all statistical comparisons are between day-matched experimental groups; see figure legend for details). To exclude the possibility that this effect was solely due to Cre overexpression, we expressed Cre pan-neuronally in wild-type animals and found that this did not affect quiescence behavior (Fig S3F). Together, these data suggest that *flp-13* expression in neurons inhibits the onset of quiescence and death in PA14-infected animals.

We next mapped out the neurons where *flp-13* is required to delay quiescence and death. We observed mScarlet expression in 14 neurons across the head region of the animal, with the highest expression in the I5 and ALA neurons (Fig S3E), and lower expression in neurons whose positions matched sites of expression according to CeNGEN, which are mostly pharyngeal neurons and non-pharyngeal sensory neurons. The same number of neurons showed *flp-13* expression in PA14-infected animals, with I5 and ALA again showing the highest levels of expression (Fig. S3E). To map out the neurons where *flp-13* expression is functionally required for its PA14-induced phenotypes, we first expressed Cre under broad genetic drivers to knock out *flp-13* in different subgroups of neurons. Loss of *flp-13* from either non-pharyngeal sensory neurons or pharyngeal neurons was sufficient to recapitulate the null phenotype (Fig. 2D). In contrast, loss of *flp-13* from ALA did not lead to any difference compared to wild-type animals and to Cre-negative controls with intact *flp-13* expression (Fig. 2D, Fig S3F). These results suggest that *flp-13* functions in pharyngeal neurons and non-pharyngeal sensory neurons, but not ALA, to prevent PA14-induced quiescence and death.

We further mapped out which specific neurons were functionally important sites of FLP-13 production using cell-specific Cre drivers. Among the pharyngeal neurons, loss of *flp-13* from I5 or I1 accelerated PA14-induced quiescence and death compared to Cre-negative controls with intact *flp-13* expression (Fig. 2E). Among the sensory neurons, loss of *flp-13* from ASH and OLL caused this same phenotype (Fig. 2F). The same sets of neurons remained functionally critical when quantifying either quiescent animals or both dead and quiescent animals together. These results identify specific non-pharyngeal sensory neurons and pharyngeal neurons that release FLP-13 to prevent PA14-induced quiescence and death.

The requirement for FLP-13 production in I5, I1, ASH, and OLL for PA14-related phenotypes contrasts with previous results showing that FLP-13 production in ALA is required for stress-induced sleep[20,22]. To confirm this functional difference, we used the same conditional knockout strains that we generated here to identify which neurons need to produce FLP-13 for stress-induced sleep. For these experiments, we used a standard stress-induced sleep assay where animals were exposed to a 30 min heat shock and the fraction of quiescent animals was quantified 30 mins later (Fig. 2G, left panel). Consistent with prior literature, deletion of *flp-13* in ALA impaired stress-induced sleep (Fig. 2G, right panel). In contrast, deletion of *flp-13* in I5 had no effect (Fig. 2G, right panel). The disruption of *flp-13* expression in ALA and I5 in each of the respective conditional knockout strains was confirmed using the mScarlet endogenous reporter (Fig S3E, bottom). Together, these results provide a clear separation between functionally required sites of FLP-13 production for stress-induced sleep versus PA14-related quiescence and death (Fig. 2H).

## DMSR-1 is a FLP-13 receptor that delays PA14-induced quiescence and death

*flp-13* has been reported to act through at least two receptors, FRPR-4 and DMSR-1, to control other aspects of behavior[55,56]. We examined PA14-induced behavioral changes in null mutants lacking each of these receptors. Whereas *frpr-4* mutants displayed a normal PA14 feeding decrease but no accelerated onset of quiescence, *dmsr-1* mutants displayed robust quiescence within 20 h of PA14 infection, matching

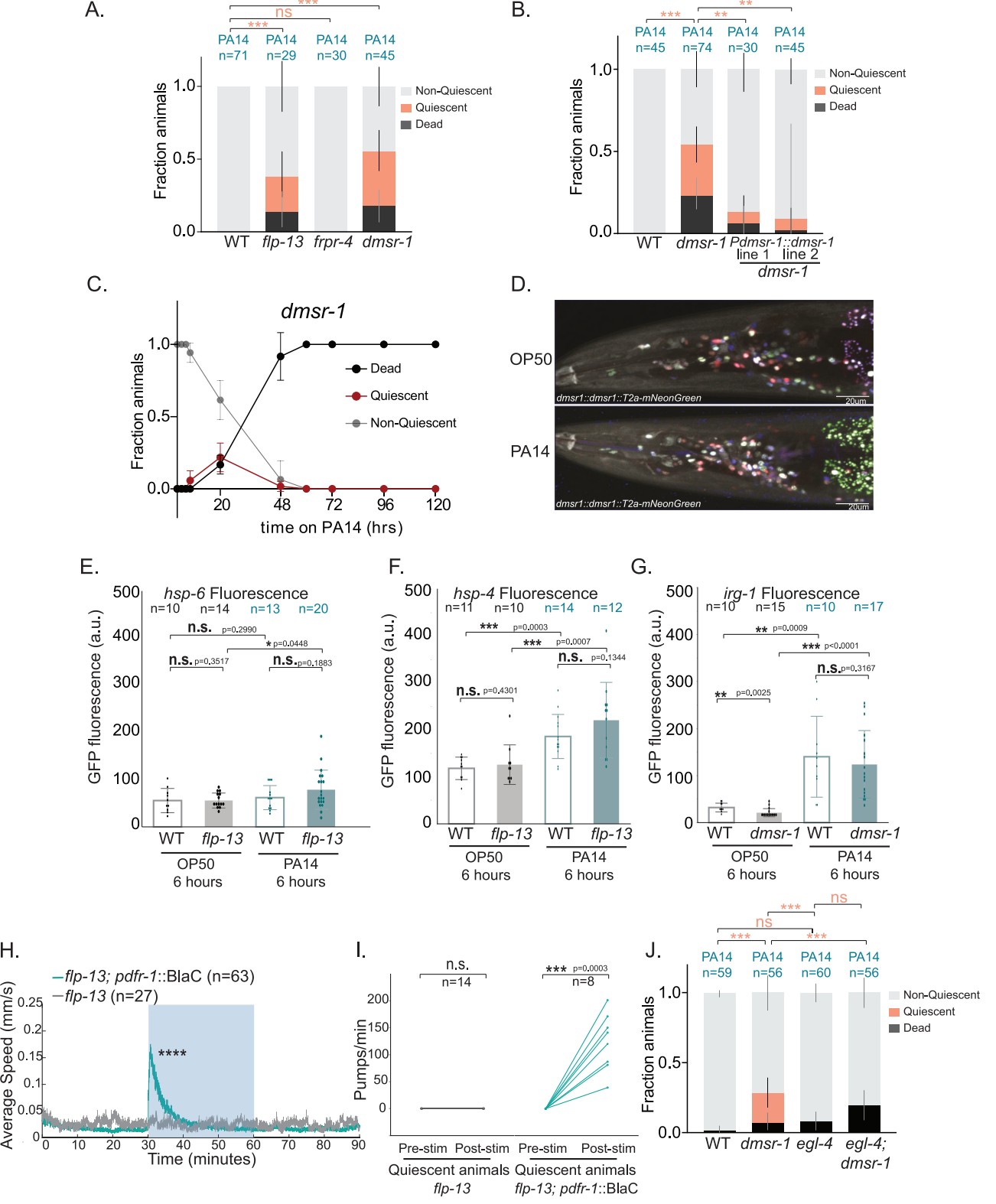

*flp-13* mutants (Fig. 3A, Fig S4A). This *dmsr-1* mutant phenotype could be rescued by restoring *dmsr-1* expression with a genomic DNA fragment containing the full *dmsr-1* gene (Fig. 3B). In time course assays, no quiescence was observed in *dmsr-1* mutants placed on OP50 for the duration of the assay (Fig S4B). However, upon infection, the time course of quiescence and death in *dmsr-1* was even more accelerated than the *flp-13* phenotype (Fig. 3C). This suggests that *flp-13* may act through the *dmsr-1* receptor to prevent quiescence and death in

PA14-infected animals. DMSR-1 is a GPCR that has the highest binding affinity to FLP-13 but also has other ligands[57,58]. The stronger phenotype of *dmsr-1* mutants, compared to *flp-13* mutants, might indicate the presence of an additional ligand that functions in parallel to FLP-13.

We next tested whether PA14 infection impacts the expression pattern of *dmsr-1*. To do so, we generated a *dmsr-1* reporter strain where t2a-mNeonGreen was inserted into the endogenous *dmsr-1* gene just before the stop codon. This strain was crossed to NeuroPAL[59] to

**Fig. 3 | Mutants lacking *flp-13* and its receptor *dmsr-1* display reversible quiescence upon PA14 infection. A** PA14-induced quiescence and death in *frpr-4(ok2376)* and *dmsr-1(qn45)* after 20 h of infection. **B** PA14-induced quiescence and death in indicated genotypes after 20 h of infection. **C** Time course of *dmsr*-1 behavior and viability, displayed as in (2 A). Error bars indicate standard deviation. For each genotype, n = 10 animals per replicate, 6 replicates. **p < 0.005, two-tailed Mann−Whitney test: Median quiescence observation was 20 hrs in *dmsr-1* vs 60 hrs in WT animals (Fig. 1A). *p < 0.05, two-tailed Mann−Whitney test: Median time point at which 50% animals died was 48 hrs in *dmsr-1* mutants vs 72 h in WT (Fig. 1A). **D** Representative images of expression of *dmsr-1* transgene (*syb6591*) overlaid with NeuroPAL after exposure to OP50 or PA14 for 20 h. No differences were found. See Fig S3 for quantification. Quantification of fluorescent intensities of (**E**) *hsp-6::GFP* (**F**) *hsp-4::GFP*, or (**G**) *irg-1::GFP*, measured 6 h after being placed on indicated bacteria. n are individual animals and are indicated on the figure. *p < 0.05, ** p < 0.01, ***p < 0.001, n.s. indicates not significant (p > 0.05) with Wilcoxon rank

sum test. Error bars indicate standard deviation. **H** Locomotion speed while blue light (0.5 mW) was applied to activate *pdfr-1::*BlaC. *flp-13* animals lacking *pdfr-1::*BlaC provide a control for the generic effects of blue light on behavior. n = 27-63 tracked animals. Wilcoxon rank sum test (p < 0.0001) was run on n = 17 (*flp-13;pdfr::BlaC*), and n = 19 (*flp-13*) tracked animals in the 5 min period immediate after blue light onset. **I** Pumping rate of quiescent animals before and after blue light (0.5 mW) exposure for 10 s. n's are individual animals and are indicated on the figure.***p < 0.001, Paired t test. **J** PA14-induced quiescence and death in indicated genotypes after 20 h of infection. For (**A**), (**B**), and (**J**) error bars represent the 95% confidence interval for the bootstrapped mean fraction values of animals. Shown data was collected across multiple days but statistical comparisons were computed between day-matched experimental groups. ***p < 0.001, **p < 0.01, fraction quiescence compared via Chi-square test with Bonferroni correction. Source data are provided as a source data file.

facilitate neuron identification. The NeuroPAL transgene has three fluorescent proteins (BFP, OFP, mNeptune) expressed under well-defined genetic drivers, which makes it easy to determine the identities of imaged neurons. We observed expression of *dmsr-1* expression in many neurons and reliably identified ~43 neurons per animal with bright reporter expression (Fig. 3D, Fig S4C–E). We then compared this set of neurons in control and PA14-infected animals but did not detect any differences (Fig. S4C–E). This suggests that PA14 infection does not change the set of neurons that prominently express *dmsr-1*.

### *flp-13* is not required for PA14-induced stress responses, and delays quiescence and death through a pathway that is independent of known immune pathways

The accelerated onset of quiescence and death in *flp-13* animals raised the possibility that these animals may have aberrant responses to infection and stress. PA14 infection is known to induce a range of stress responses, which can be measured by quantifying induction of responsive reporter genes: mitochondrial damage response (*hsp-6* reporter)[60], unfolded protein response (*hsp-4* reporter)[61], and innate immune responses (*irg-1* reporter)[62] (Fig. 3E–G). We examined induction of these reporters in wild-type and either *flp-13* or *dmsr-1* mutants, based on which gene was unlinked to the integrated stress reporter gene, to ease strain construction. We focused on an early timepoint (6 h) before there might be indirect effects due to animals transitioning into quiescence. *hsp-4* and *irg-1* were reliably induced at 6 h, with no detectable difference between wild-type and *flp-13/dmsr-1* (Fig. 3F, G). *hsp-6* induction was only observed at later timepoints (Fig S5A), with no difference observed in baseline expression at 6 h in wild-type and *flp-13* (Fig. 3E). All three reporters showed increased expression at later stages of infection (20 h) in both WT and *flp-13/dmsr*-1 animals (Fig S5A). *flp-13* animals, which exhibit quiescence at this time point, had elevated expression of *hsp-4* and *hsp-6*. These results suggest that several stress response pathways are activated normally in early stages of infection in *flp-13/dmsr-1* mutants. Related to this, we also noted that *flp-13* animals appear morphologically similar to wild-type animals after 20 h of PA14 infection, unlike innate immunity-defective *pmk-1* animals, which appear visibly sick at this timepoint (Fig S5B).

We also examined interactions between *flp-13* and the genetic pathways that have been shown to delay PA14-induced death in *C. elegans*, namely the innate immune pathways. Mutants lacking the immune effectors *fshr-1*[63] and *sek-1*[39], which are key members of two independent immune pathways, displayed accelerated quiescence and death (Fig. S5C). Deletion of *flp-13* in each of these backgrounds led to a stronger, additive phenotype with even further accelerated quiescence and death (Fig. S5C). This suggests that defects in immunity can cause early quiescence and death, but that *flp-13* does not function together in a linear pathway with these two well-characterized immune pathways. Mutants lacking the mitochondrial stress regulator *atfs-1*[64] or the immune regulator *zip-2*, which are both also linked to PA14 immunity,

did not display accelerated PA14-induced quiescence, and deletion of *flp-13* in these mutant backgrounds yielded phenotypes matching *flp-13* single mutants (Fig. S5C). This suggests that defects in mitochondrial stress and the *zip-2* immune pathway do not impact chronic PA14-induced quiescence or the *flp-13* mutant phenotype. Together, these data suggest that *flp-13* functions independently of known PA14 innate immunity pathways to delay quiescence and death.

### PA14-induced quiescence is a reversible quiescence state with no change in arousal threshold

Our analysis here of *flp-13* mutants revealed a form of quiescence in PA14-infected animals that had not been characterized before. Thus, we next examined whether the PA14-induced quiescence state has properties that match other forms of quiescence in *C. elegans*: reversibility, similar genetic requirements, and increased arousal threshold (Fig. 3H-J; Fig. S5D, E). First, we examined whether this form of quiescence was reversible by testing whether quiescent *flp-13* animals could be acutely aroused out of quiescence. For this, we drove expression of the light-activated adenylyl cyclase BlaC under the *pdfr-1* promoter (*pdfr-1::*BlaC) in *flp-13* mutant animals. Activation of *pdfr-1::*BlaC with blue light has been previously shown to induce robust roaming states[65], which are sustained bouts of high arousal locomotion (Fig. 3H). We examined how quiescent, PA14-infected *flp-13; pdfr-1::BlaC* animals responded to BlaC activation via blue light illumination. This led to an immediate onset of roaming behavior, peaking at an average speed of 0.15 mm/sec, matching typical fast roaming speeds[24,65]. Quiescent *flp-13; pdfr-1::BlaC* animals also began feeding upon blue light stimulation (Fig. 3I). This indicates that *flp-13* quiescence is reversible: animals can still exhibit robust locomotion and feeding when optogenetic interventions acutely change the arousal state of their nervous system.

Null mutations in *egl-4*/cGMP-dependent protein kinase (PKG) reduce all known forms of quiescence in *C. elegans*, potentially because these mutations result in high levels of sensory arousal[28,47,66]. To test whether PA14-induced quiescence in *flp-13* or *dmsr-1* animals was also dependent on *egl-4*, we examined PA14 responses in *dmsr-1;egl-4* double mutants. We chose *dmsr-1*, since this gene is unlinked to *egl-4* and eased double mutant construction. Indeed, these animals did not display accelerated PA14-induced quiescence after 20 h of infection (Fig. 3J). However, the elevated PA14-induced death rates observed in *dmsr-1* mutants were still observed in *dmsr-1;egl-4* double mutants (Fig. 3J). This result suggests that the death phenotype in *dmsr-1* mutants is not strictly dependent on these mutants displaying quiescence. We also examined the impact of the *egl-4* mutation in an otherwise wild-type background. Here, we assayed animals after 48 h of infection, a timepoint when WT animals begin to display quiescence. Again, we found that this quiescence was attenuated by the *egl-4* mutation (Fig S5E). This suggests that PA14-induced quiescence requires *egl-4*. Taken together, these results suggest that *flp-13* and

*dmsr-1* mutants exhibit accelerated PA14-induced quiescence, which is a reversible behavioral state that requires *egl-4*, like other forms of quiescence in *C. elegans*.

We also tested whether these quiescent animals have an altered arousal threshold. For this, we used a standard assay where animals are exposed to aversive blue light and the latency to respond with reverse movement is quantified. We examined wild-type and *flp-13* animals before and after PA14 infection. Infected *flp-13* mutants that were quiescent after 20 h of PA14 exposure could still respond to blue light reliably and did not have the increased arousal threshold seen in other forms of quiescence (Fig S5D). These results suggest that PA14-induced quiescence does not include an increased arousal threshold, distinguishing it from other forms of quiescence. The suggests that PA14-induced quiescence may be more akin to lethargy than sleep.

Finally, to disentangle quiescence from a stupor-like state before death, we performed a longitudinal analysis of individual *flp-13* animals from first exposure to PA14 until their death several days later (Fig S6A). At each timepoint, we quantified the fraction of animals that were normal, quiescent, or dead. We observed that approximately half the animals showed transitions from normal behavior to quiescence and then reversed back to normal behavior. A similar fraction of animals transitioned from normal to quiescent states and died without any observed transition back to normal behavior. A small fraction of animals died without showing observable quiescence. These qualitative observations further suggest that while infection-induced quiescence and death are correlated phenotypes, quiescence is not a strict predecessor to death. This quiescence state can be reversed by acute stimulation (Fig. 3I) and can reverse back to normalcy on its own accord (Fig. S6A).

### The PA14-induced quiescence state is mediated by ASI activation and *daf-7*

We next sought to determine the neural mechanism that drives PA14-induced quiescence. To do so, we performed brain-wide calcium imaging in infected *flp-13* animals as they entered and exited quiescence 20 h after infection. We performed these studies in freely-moving *flp-13* mutant animals co-expressing pan-neuronal NLS-GCaMP7f and the NeuroPAL transgene to facilitate neuronal identification. Infected animals exhibiting quiescence were selected and were exposed to PA14 during 8-min of freely-moving calcium imaging sessions (Fig. 4A). At the end of each imaging session, they were immobilized by cooling and multi-spectral images were captured for NeuroPAL annotations. The live-tracking microscope, software for extracting calcium and behavioral data, and procedure for neuron annotation have been previously described and validated[67].

We recorded data from eight PA14-infected animals that displayed bouts of quiescence during the recordings. We measured several behavioral variables including feeding rates, locomotion speed, and head curvature, and used these metrics to identify quiescent bouts where there was no movement or feeding (Fig. 4B). We recorded 117-176 neurons per animal and determined the identities of 40–121 of these neurons (Fig. 4B, C). Bouts of quiescence (quantified as described above) were accompanied by a significant decrease in activity in several neuron classes known to encode locomotion (RMD, AVA, OLQ, and others) and feeding (MC, M3, etc.) behaviors (Fig. 4D). A smaller subset of neurons displayed elevated activity during quiescence bouts. Other neurons showed no GCaMP changes throughout recordings, which could reflect inactivity or low activity indiscernible from experimental noise. To determine which neuron classes displayed elevated activity during quiescence, we quantified each neuron class's activity change during quiescence. The neuron classes with the largest activity increase during quiescence were AVB, I4, AVL, ASI, M2 and RIH (Fig. 4C). ASI has been previously shown to be functionally required for satiety-induced quiescence[28,68], making it an attractive candidate to mediate quiescence in this context. ASI displayed higher activity

during quiescence in two of the three datasets where we could identify the neuron (Fig. 4E). These results raise the possibility that ASI activity increases during PA14-induced quiescence, at a time when many behavioral circuits display diminished activity.

To confirm these results in a larger group of animals, we generated a strain with cell-specific expression of GCaMP7f in ASI (under the *srg-47* promoter) and recorded ASI activity in 17 additional PA14-infected animals. Consistent with the above results, ASI activity was correlated with quiescence bouts in these recordings (Fig. 4F). These data indicate that ASI activity is elevated during PA14-induced quiescence in *flp-13* mutants, though we note that its activity is not perfectly predictive of PA14-induced quiescence.

We next examined which quiescence pathways functionally contribute to PA14-induced quiescence (Fig. 4G–J). To test whether ASI mediates PA14-induced quiescence, we ablated ASI in a *flp-13* mutant background using a previously characterized ASI::Caspase strain[69]. The loss of ASI delayed the onset of PA14-induced quiescence in *flp-13* animals (Fig. 4G, J left panel, Fig S7A–D), suggesting that ASI contributes to PA14-induced quiescence in these animals. Since ASI does not express *flp-13*, but does express *dmsr-1*[70], FLP-13 could potentially inhibit ASI to delay infection-induced quiescence. However, the additive nature of the *flp-13* and ASI- phenotypes (Fig. 4G) suggests the presence of other factors.

We also examined whether other neurons and genes that control *C. elegans* quiescence in other contexts impact PA14-induced accelerated quiescence in *flp-13* mutants. We found that loss of RIS (via *aptf-1* mutation[71]), or the *nlp-22* neuropeptide-encoding gene (which promotes quiescence[72]) did not alter PA14-induced quiescence in *flp-13* mutants (Fig. 4H). Since loss of ALA (via *ceh-17* mutation) is known to suppress quiescence in WT animals, we tested if it also suppresses quiescence that appears earlier in *flp-13* mutants. Interestingly, loss of ALA dramatically increased the death rate in *flp-13* mutants, indicating that the *ceh-17* mutation is a genetic enhancer of the *flp-13* mutation. Overall, our results suggest that PA14-induced quiescence in *flp-13* mutants depends in part on the ASI neuron, which has elevated activity during this state.

ASI is known to release the secreted peptide DAF-7/TGF-beta, which impacts many aspects of *C. elegans* behavior and physiology[24,73–75]. Previous work showed that *daf-7* expression in ASI is increased upon PA14 infection[32]. In addition, other studies showed that *daf-7* expression in ASI is important for satiety quiescence in fasted animals re-introduced to food. Moreover, *daf-7* is also required for movement quiescence in well-fed animals exposed to noxious stressors such as heat shock or UV exposure[76,77]. We tested whether *daf-7* was required for the quiescence observed in PA14-infected *flp-13* mutants by examining *flp-13;daf-7* double mutant animals. Strikingly, *flp-13;daf-7* double mutants displayed no quiescence or death after 20 h of infection and attenuated quiescence and death at later time points (Fig. 4I, J right panel), revealing a strong suppression of the *flp-13* phenotype. We also examined PA14-induced quiescence in *daf-7* single mutants. In contrast to wild-type animals, *daf-7* single mutants did not exhibit quiescence three days after infection (Fig. 4J right panel) and showed survival to later time points as well (Fig S7E). The intermediate nature of the *daf-7;flp-13* double mutant phenotype suggests that additional PA14-induced quiescence pathways contribute to quiescence in *flp-13* mutants as well (Fig. 4I-J, Fig S7A, B, E, F). Overall, these results suggest a significant function for *daf-7* in PA14-induced quiescence and in the accelerated quiescence phenotype of *flp-13*.

## Discussion

Neuromodulators play pivotal roles in coupling changes in animals' internal states to their behavior. How combinations of neuromodulators released from different neuronal sources control the diverse internal states that animals exhibit remains an open question. In this

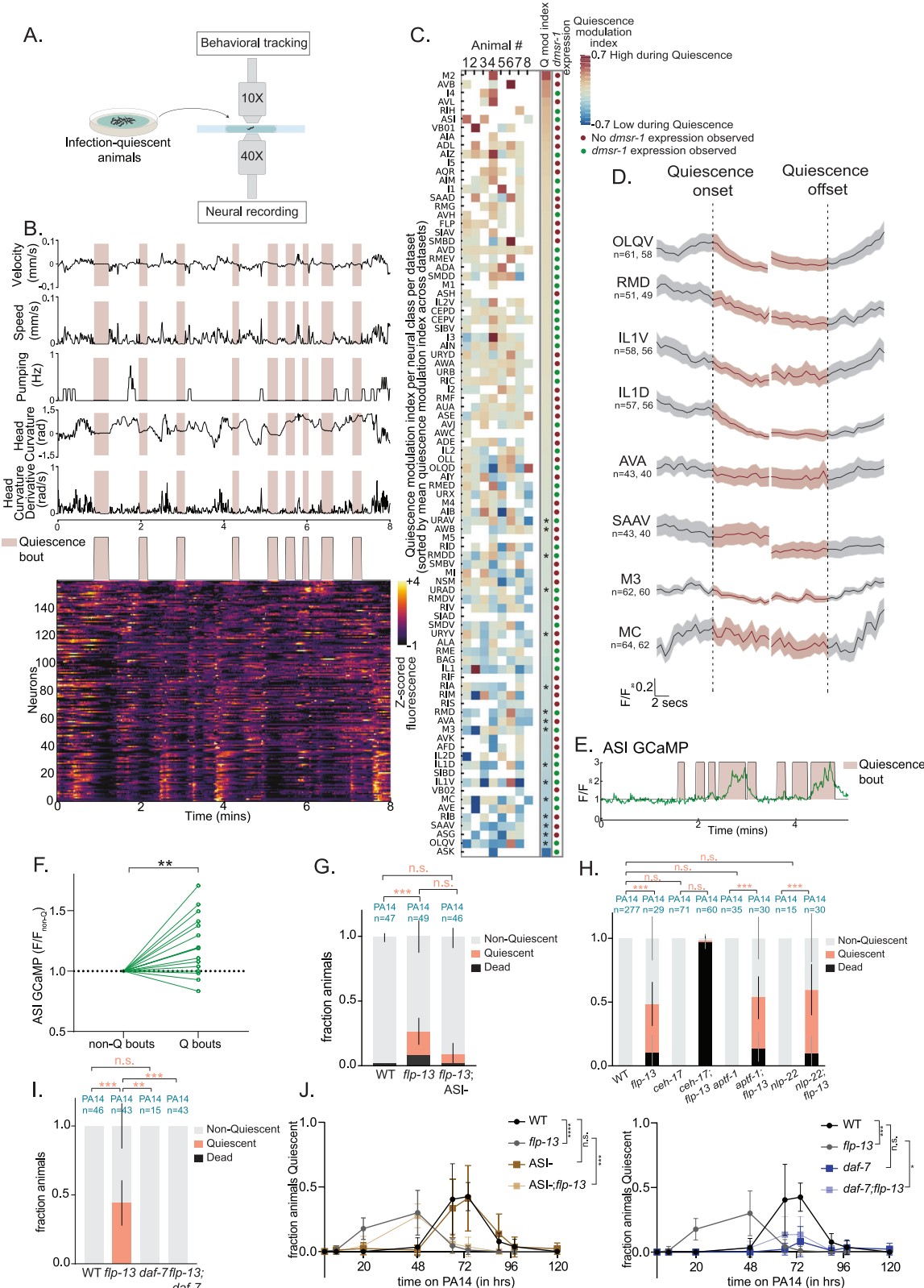

study, we identified neuromodulatory pathways that control the behavioral changes observed in pathogen-infected *C. elegans*. In animals that have recently become infected by PA14, feeding behavior is reduced by the release of neuropeptides from ALA, a pathway that is also known to induce stress-induced sleep in uninfected animals. In infected animals, one ALA neuropeptide, FLP-13, takes on a different role and becomes critical for viability. This function of FLP-13 is

mediated by its release from non-pharyngeal sensory neurons and pharyngeal neurons, rather than ALA. The onset of PA14-induced quiescence that is observed in severely infected animals requires ASI and DAF-7/TGF-beta, which also control satiety-induced quiescence in uninfected animals. Our findings show that the neuromodulators that control PA14 infection-induced behavioral changes overlap with those that control behavioral states associated with stress and satiety (Fig. 5).

**Fig. 4 | Brain-wide imaging identifies ASI and DAF-7/TGF-Beta as critical components of infection-induced quiescence. A** Experimental setup. **B** (Top) Behavior of an infected *flp-13* animal. Quiescence (pink) bouts were periods of >8 s when speed<0.01 mm/s, head curvature derivative<0.1 rad/s and pumping=0 Hz. (Bottom) GCaMP signals of 159 neurons in the same animal. **C** Heatmap summarizing modulation of neural activity by quiescence. Rows represent neuron classes and columns represent individual animals. Quiescence modulation index indicates the change in mean activity during quiescent bouts vs mean activity outside quiescent bouts, relative to the range of activity for each neuron; *p < 0.05, two-tailed empirical P value with multiple correction (See "Methods"). *dmsr-1* expression is from Fig. 3D and S4. **D** Event triggered averaged activity traces of motor and pharyngeal neural classes during quiescence bouts. Error bands are standard deviation. **E** Neural activity of ASI, from a single infected *flp-13* animal from the brain-wide imaging dataset. **F** Normalized ASI GCaMP7f signal in PA14-infected *flp-13* mutants. Each dot indicates mean GCaMP signal during non-quiescence vs quiescence bouts for a single animal; lines connect dots coming from same animals.

n = 17 animals. **p < 0.01, two-tailed Wilcoxon matched-pairs signed rank test. **G** PA14-induced quiescence and death in indicated genotypes. An ASI::Caspase strain was used. **H** PA14-induced quiescence and death in indicated genotypes. Alleles used include *ceh-17(np1)*, *aptf-1(tm3287)*, *nlp-22(gk509904)* and *flp-13(tm2427)*. **I** PA14-induced quiescence and death in indicated genotypes. Alleles used include *daf-7(e1372ts)* and *flp-13(tm2427)*. For (**G–I**), error bars are 95% confidence interval for the bootstrapped mean fraction of animals that were quiescent, non-quiescent, and dead, respectively. Shown data was collected across multiple days but statistical comparisons were computed between day-matched experimental groups. ***p < 0.001,**p < 0.01, n.s. indicates not significant, Chi-square test with Bonferroni correction. **J** Time course of animal quiescence during infection. Error bars indicate standard deviation. For each genotype, 10 animals were placed on a single plate; three replicates with three plates each were performed. *p < 0.05, ***p < 0.001, n.s., p > 0.05. Median quiescence values compared via two-tailed Mann–Whitney test. Error bar indicates standard deviation. Source data are provided as a source data file.

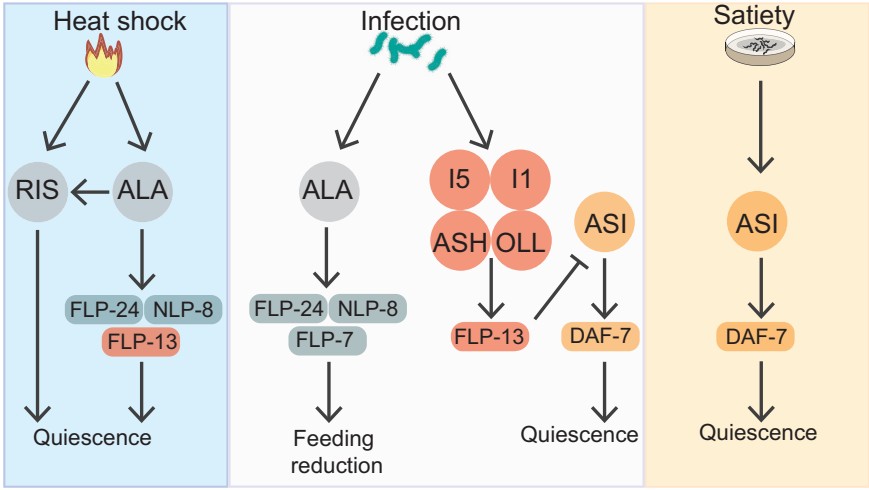

**Fig. 5 | Model for PA14-infection-induced behavioral changes in *C. elegans*.** Neuromodulatory pathways associated with stress (blue panel, left) and satiety (yellow panel, right) are recruited upon infection to drive changes in feeding and quiescence upon infection. Upon heat shock, the ALA neuron induces quiescence through the action of its neuropeptides, including the FMRFamide peptide FLP-13 (red rectangle)[20,22] and via the neuron RIS[48,49]. In satiated animals, the neuromodulator TGF-beta/DAF-7 is released by the ASI neurons to promote quiescence. These neuromodulatory pathways are recruited in the infection context (center panel). While ALA continues to suppress behaviors like feeding, FLP-13 delays quiescence and is released from other neurons (red circles). The effects of FLP-13 to delay quiescence are suppressed by the quiescence-promoting ASI and TGF-beta/DAF-7 pathway.

This suggests that the states of stress, satiety, and infection are not induced by unique sets of neuromodulators. Instead, one larger set of neuromodulators may be deployed from different sources and in different combinations to specify these different internal states.

### Infection reconfigures the pathways involved in stress-induced sleep

When uninfected animals are exposed to stressors such as heat-shock or ultraviolet irradiation, the ALA neuron promotes stress-induced sleep by suppressing a suite of behaviors[20,22,47], including feeding. Our work extends this role to the context of infection, where ALA suppresses feeding behavior in infected animals. ALA expresses many neuropeptide genes, including *flp-7* as well as *flp-24, nlp-8*, and *flp-13*, which collectively induce quiescence. We found that the neuropeptides FLP-24, NLP-8 and FLP-7 are required for feeding reduction upon PA14 infection. These neuropeptides may either be released by ALA or by other co-expressing neurons with infection-specific roles. Interestingly, upon heat shock, *flp-24* and *nlp-8* suppress locomotion and defecation but not feeding[20], while *flp-7* does not have roles in this context[20]. However, upon infection, each neuropeptide is required for the feeding reduction. Therefore, while two of these neuropeptides continue to suppress behaviors across contexts, all exhibit different roles in specifically suppressing feeding upon infection. The

interactions between the peptides also appear different, as they have redundant requirements after heat-shock, but non-redundant functions after infection. Additional experiments are required to unravel these context-specific functions of *flp-24, nlp-8* and *flp-7*.

Like the other ALA neuropeptides, FLP-13 suppresses behaviors to promote quiescence in uninfected animals exposed to stress[20,22,47,77]. In PA14-infected animals, however, we find that FLP-13 instead delays quiescence and death, a phenotype that is typically classified as an innate immune function. We identified the pharyngeal I1 and I5 neurons, and the non-pharyngeal sensory ASH and OLL neurons, but not ALA, as critical sites of infection-induced FLP-13 function. It is currently unclear why these specific neurons play a role since we were unable to identify evident changes in the expression patterns of *flp-13* upon infection. One explanation might have been a dosage-dependent effect of *flp-13* where neurons expressing higher levels of *flp-13* play a more critical role in its functions. However, our results are not fully consistent with this explanation: I5 and ALA display the highest *flp-13* expression levels in uninfected and infected animals but disrupting *flp-13* in ALA did not impact infection-induced quiescence and deleting *flp-13* in I5 did not impact heat shock-induced quiescence. Thus, the functional relevance of the neural source does not fully map onto neurons with the highest *flp-13* transcriptional expression. Further, infection did not alter which cells express the *flp-13* receptor encoding gene, *dmsr-1*.

Other untested hypotheses may explain why specific pharyngeal and non-pharyngeal sensory neurons, and not ALA, become critical FLP-13 sources upon infection. One possibility is that chronic infection may increase I5, I1, ASH and OLL activity, resulting in increased FLP-13 release by these neurons. Alternatively, the I1, I5, ASH and OLL neurons may release co-transmitters with FLP-13 that modulate FLP-13 function. Such co-release and co-transmission modulation is pervasive, evolutionary conserved, and lends greater flexibility to neural circuits. The neuron I5 is known to inhibit the inhibitory motor neuron M3[78]. One possibility is that altered function of I5 upon infection leads to hyper-activation of M3 and consequent pumping inhibition. Additional studies will be necessary to determine the mechanism at work here, as well as the complex interactions between neuromodulators that give rise to different internal states. Similar complexities are likely to exist in vertebrate systems where internal states are best decoded from combinations of neuromodulator-producing cell types, rather than individual cells types[79,80].

### Distinct mechanisms underlie different forms of behavioral quiescence

Reversible behavioral quiescence is a conserved component of animal behavior, from jellyfish[81] to humans. The reversibility of quiescence differentiates it from immobile states such as paralysis, coma or death. Diverse environmental and homeostatic cues induce quiescence, which despite being behaviorally homogenous can be caused by distinct neural sub-programs to serve different functions. For instance, in reptiles, birds and mammals, behavioral quiescence is observed during both rapid eye movement sleep (REM) and non-REM (NREM) sleep[82–84]. However, REM and NREM have distinguishable brain activity patterns[85] and functions in memory consolidation and the regulation of emotion[86,87]. In *Drosophila*, quiescence observed in response to sleep deprivation versus that during natural sleep cycles are mechanistically and functionally distinct[88,89]. Similarly, in *C. elegans*, the neural circuits that generate quiescence during developmentally timed sleep at larval transitions[66,72], stress-induced sleep in adults[20,22,47] or upon satiation[68], are largely separate[90].

Our study characterizes a previously uncharacterized form of quiescence in adult *C. elegans* that is displayed upon PA14 infection, is reversible, and is accompanied by reduced activity in many neuronal cell types that encode behavior. However, in contrast to other forms of quiescence, it does not correlate with an increased arousal threshold, suggesting that it may be more akin to lethargy than sleep (the exact terminology used to describe this quiescence state may need to be revisited in the future once it is more completely understood). We did not observe any involvement of ALA and RIS in infection-induced quiescence, even though these neurons are critical for *C. elegans* quiescence in other contexts (Fig. 4C). However, we did detect an important role for ASI and DAF-7/TGF-beta in infection-induced quiescence; this pathway has been linked to satiety quiescence previously. The involvement of different neurons and modulators in these different quiescent states could be due to the different timescales of the inducing stimulus – for example, whether they are acute versus chronic stressors – or different molecular pathways that are engaged by these different stimuli. Whether these different forms of quiescence facilitate different aspects of homeostatic physiology remains to be understood. Interestingly, the onset of PA14-induced quiescence is often followed by death. Whether infection-induced quiescence functions as a last resort for homeostatic recovery in infected animals remains to be explored.

## Methods

### Plasmids

The pSM-nCre expression plasmid was previously described[91]. Promoters used for nCre expression were: *ceh-45* (2.5 kB), *ceh-53* (1.5 kB), *tax-4*[92], *osm-6* (2.4 kB), *trh-1* (2.0 kB), *ceh-34* (3.7 kB), *sra-6* (3.8 kB), *ser-2d* [93], *ceh-28* (650 bp), *sra-9*[92], *gcy-33*, *tag-168*, and *gcy-36*[92].

The *dmsr-1* genomic rescue plasmid was constructed by inserting a region of *dmsr-1* genomic locus (spanning from 3.4 kB upstream of start codon up to the stop codon; including introns and exons) into pSM-t2a-GFP.

### New alleles generated in this study

Detailed information about all strains used in this study are available in Table S1.

The *flp-13* conditional knockout allele was generated through two iterative CRISPR/Cas9 genome editing steps. In one step, a loxP site was inserted 62 bp after exon 1 of the gene. In a second step, a t2a-mScarlet-loxP sequence was inserted immediately before the stop codon of *flp-13*.

We constructed two *dmsr-1* fluorescent reporter strains via CRISPR/Cas9 genome editing. For the *dmsr-1a* reporter, a t2a-mNeonGreen sequence was inserted immediately before the stop codon of the *dmsr-1a* isoform. For the *dmsr-1b*, the same strategy was used, inserting t2a-mNeonGreen immediately before the *dmsr-1b* stop codon. We failed to observe any fluorescence in the *dmsr-1b* reporter line. The images reported here are exclusively from the *dmsr-1a* reporter.

### Preparation of bacteria for experiments

*E. coli* OP50 was grown overnight in Luria Broth (LB) cultures at 37 °C under shaking conditions. *Pseudomonas aeruginosa* strains (PA14 and other mutants) and *Serratia* Db11 single colonies were inoculated in 6-8 ml of LB in a 15 ml round bottom tube with loose cap to allow for adequate aeration and grown at 37 °C under shaking conditions for 16 h. All bacteria were stored at 4 °C and used within one week.

UV-killing was performed by exposing plates seeded with PA14 to a UV transilluminator (302 nm) for 30 min. The exposure time needed to kill bacteria depended on the amount of bacteria seeded on the plate. For a 200 µl lawn grown for 20 h at room temperature (RT, ~22 °C), 30 min was sufficient for bacterial killing. This was verified by culturing bacteria from random zones on the plate to check for overnight growth at 37 °C. Plates were kept facing the UV light source with lids off. After exposure, plates were left on the bench to come back to RT before adding animals.

### Behavior Experiments

All behavioral experiments were done over at least two days and over multiple experimental replicates. All animals were added onto experimental bacteria as adults for specific durations of time. All animals were staged as L4s the day before the experiment.

### Feeding and quiescence assays

Low peptone NGM plates (3 g NaCl, 22 g agar, 0.75 g peptone, 1 ml cholesterol (5 mg/ml) per 1 L media) were used as experimental plates across all experiments to slow down growth of bacteria. 200ul of OP50 or PA14 were seeded on 10 cm low peptone NGM plates and spread using a bacterial spreader to make a large lawn which did not completely cover the plate. Plates were left at RT for 20 h stacked horizontally. Five animals were added per plate per bacterial condition and left undisturbed in a box inside a 22°C incubator for 20 h after which feeding/pumping behavior was measured. Grinder movements or pharyngeal pumps were counted for each animal with a manual tally counter (Digi 1st TC-890 Digital Tally Counter) under a benchtop microscope for 20 s. Pharyngeal pump counts are multiplied by 3 and plotted as pumps/min across figures. Standard pumping experiments were performed at least on 2 different days with 3 plates per condition (15 animals) for each strain/genotype. While some early initial pilot experiments were performed unblinded, all subsequent manual

pumping quantification was performed blinded to genotype and all key experiments were re-confirmed under blinded quantification.

Quiescence assays were performed using the same plates described above. An animal showing complete lack of pumping, body movement or locomotion for the 20 s of observation time was marked as quiescent. Animals that showed only partial phenotypes, that is, were only pumping quiescent or lacked movement in the 20 s of observation, were categorized as "non-quiescent". Most animals that were pumping quiescent were also locomotion quiescent. For example, in Fig. 2C, for *flp-13(tm2427)*, 9% animals were dead, 42% animals were completely quiescent (and categorized as quiescent), 2% animals were pumping quiescent only, 16% animals were locomotion quiescent only, and 31% did not show any form of quiescence. The last three categories were together categorized as "non-quiescent" (49%). Plates were handled gently so as to not to rouse any potential quiescent animals. Animals that did not respond at all after poking with a platinum wire pick were considered as dead. For Fig S6, the same experiment was performed but with a single animal on each plate.

For experiments with the *flp-13;pdfr-1::BlaC* strain, 0.5 mW blue light (stage backlight) was used on the Leica Fluocombi III stereomicroscope using the 5X objective to quantify feeding rates. BlaC is very sensitive to blue light and hence a filter was placed on the microscope stage with the stage backlight to prevent blue light from the backlight from stimulating the animals at baseline. The filter was carefully removed from under the experimental plate taking care not to mechanically disturb the animals. Feeding was quantified for 20 s as in previous experiments immediately after exposure to the blue light.

### Survival Assays
Low peptone NGM plates were seeded with 200ul of bacteria same as above. 10 animals were added per plate, and number of animals that were quiescent (as defined previously) or dead were measured at specific time points (0 h to 120 h). Animals were moved to plates with equivalent lawns of bacteria (seeded at the same time) every 2 days to ensure that the animals were not confused with their adult progenies.

### Pumping with exogenous serotonin
Animals were exposed to PA14 for 20 h as described above. Feeding was quantified after 20 h of infection. These animals were then transferred to 6 cm low peptone plates with added 15 mM serotonin (Sigma Aldrich- H7752; 50 mM serotonin stocks were made in water and added to molten agar right before pouring) for 10 mins, and pumping was quantified as above. Serotonin plates were made the day before the experiment, seeded with 50 µl OP50 and kept at room temperature.

### *flp-13* and *dmsr-1* reporter imaging
Animals were added on OP50 or PA14 low peptone plates for 20 h as described above and imaged using a Zeiss LSM900 confocal microscope system. Animals were immobilized on a flat agar pad using 100uM sodium azide (Sigma Aldrich). For *flp-13::mScarlet* expression, all neurons expressing *flp-13* were observed using a Z stack, and maximum intensity projections were used to observe all neurons in one frame. For the *dmsr-1::T2A-mNeonGreen* imaging, all neurons expressing *dmsr-1* were overlaid with NeuroPAL expression. The atlas from Yemini et al.[59] was referred to for neuronal identification. A subset of neurons with the highest *dmsr-1* expression were selected across the head region of the animal to quantify levels of expression in infected and uninfected conditions. For neuron classes that had L/R pairs, the brighter cell was chosen.

### Stress reporter imaging
Animals were placed on OP50 or PA14 low peptone plates for 6 h or 20 h as described above. Multiple animals were added onto 500ul agar pads and immobilized with 100uM sodium azide (Sigma Aldrich).

For each reporter strain, animals on OP50 and PA14 were imaged using a custom Andor spinning disk confocal system, containing a 5000 rpm Yokogawa CSU-X1 spinning disk unit and Borealis upgrade, built on a Nikon ECLIPSE Ti microscope[67] at the same laser intensity using a 10X objective (CFI Plan Fluor 10x, Nikon). Images were analyzed using Fiji[94] where a region of interest (ROI) was drawn around the region of expression, extending from the region immediately posterior to the pharyngeal bulb along the full length of the animal's body. Expression was mostly seen in different parts of the intestine for the tested strains. Mean fluorescence intensity for this ROI was calculated from which the mean fluorescence intensity of a circle drawn outside the animal was subtracted to perform background subtraction.

### Multi-animal behavioral recordings
For recordings of animal speed, multi-worm tracking recordings were performed essentially as previously described[95]. Animals were pre-exposed to bacteria as described in the text and in the bacteria methods section above. Animals were recorded on Streampix software at 3 fps. JAI SP-20000M-USB3 CMOS cameras (5120×3840, mono) with Nikon Micro-NIKKOR 55 mm f/2.8 were used for recordings. Backlighting was provided by IR LEDs (Metaphase). Data analysis was conducted using previously-described custom MATLAB scripts[95]. For optogenetic BlaC stimulation, light was supplied from a 470 nm (at 0.5 mW/mm$^2$) Mightex LED at defined times in the video. Custom Matlab scripts were used to analyze videos to derive animal speed.

### Heat shock experiments
Heat shock experiments were derived from protocols used by Nath et al. 2016[20] and Nelson et al. 2014[22]. Young adult animals were added on 200 µl OP50 lawns on 10 cm low peptone plates as described above for 20 h. For the heat shock, 10 animals were transferred to 6 cm plates which were seeded overnight with 50ul OP50 and pre-warmed in a 35°C incubator. These plates were immediately sealed with parafilm and kept immersed in a 35 °C circulating water bath for 30 min. The lids were exchanged with dry lids and plates were kept inverted to minimize condensation after taking out of the water bath and left undisturbed for 30 min. Plates were then handled very carefully to not rouse animals out of quiescence. Quiescence was then measured as described above using a 20 s observation period. Heat shock times were staggered to be able to assay individual plates exactly at 30 min after heat shock.

### Single neuron calcium imaging
Calcium imaging of ASI was conducted as previously described for other neurons[95]. Animals with ASI-specific expression of GCaMP7f were infected with PA14 for 20 h starting on their first day of adulthood, as described above. After the 20 h infection period they were placed on imaging slides. Imaging slides were prepared by placing a 3 µL drop of PA14 on a minimally thick NGM pad. A thin silicon sheet (0.01" thick; Stockwell Elastomerics) was custom cut in the shape of a corral where a rectangular arena of the agar pad was bordered on all four sides by the silicon sheet. This helped prevent the animal from leaving the pad, keeping them in the region being imaged. Five to eight animals were picked (without food) from their PA14 infection plates to the center of the corral and covered in a coverslip. Animals were allowed to acclimate for 5 mins after which they were imaged on PA14. Individual slides were imaged for no longer than 20 mins. Animals were imaged in a widefield epifluorescence configuration. Exposures alternated between GCaMP imaging and brightfield imaging, controlled by NI-DAQ triggering. The pattern of acquisitions was a frame rate of 16 fps for behavior and 4 fps for calcium signals. Exposures were a maximum of 10 ms to avoid motion blur. Data were acquired through a 4X/0.2NA objective and Andor Zyla 4.2 Plus sCMOS camera. Neurons were segmented through custom ImageJ scripts. Behavior was quantified using custom Image J (Fiji) macros and Datavyu.

## Brain-wide calcium imaging

**Mounting and recording.** Young adult SWF994 animals were added to 200 µl PA14 lawns on low peptone plates as described before. A concentrated PA14 culture to be used in the mounting buffer was grown as mentioned before (bacterial culture section) on the day before the experiment. 1 mL of the fresh PA14 culture was pelleted and then resuspended in 80 µL of M9. This was used as the mounting buffer. A thin, flat agar pad (2.5 cm × 1.8 cm × 0.8 mm) was made from low peptone NGM immediately prior to each animal being imaged. 4 µl of a concentrated solution of 100 µm microbeads in M9 was pipetted on the four corners of the pad in a single layer to maintain a gap between the pad and cover-slip so that free movement of the animal was possible. An infected quiescent animal was transferred to the center of the agar pad and 9.5ul mounting buffer was added on top. A glass cover-slip (#1.5) was then carefully positioned on top taking care to not insert any bubbles.

We started each imaging session while the infected animals were aroused in order to observe frequent transitions between quiescence and active behavior. Each recording lasted for 8 min. Animals that were quiescent for more than half of the recording were discarded since their postures during quiescence and active behavior were dramatically different, which prevented high-quality registration of neurons between timepoints. For subsequent analysis we used time segments that had at least 2 consecutive minutes of quality image registration where the animal underwent multiple switches between quiescent bouts and active behavior. Out of the eight animals, five animals (animals # 3, 4, 5, 7, and 8 in Fig. 4) had 8-min of usable time segments, one animal (animal #1) had a 5-min time segment, and 2 animals (animals #2 and 6) had 2-min time segments included. For all included time segments, we included all neural traces where we could reliably identify neurons with NeuroPAL markers with a confidence score of 3 or above. These criteria allowed us to obtain as many neural traces from as many independent animals as possible and were done prior to any neural data analysis or subsequent statistical measurements.

Since infected animals generally expressed dimmer fluorescence, we excited each fluorophore with higher laser intensities during freely-moving recordings than previously described[67]. During freely-moving recording, we excited GCaMP7f with a 488 nm laser at 14% of its maximal intensity and TagRFP (together with CyOFP and mNeptune) with a 561 nm laser at 19% of its maximal intensity, respectively. As for the RGB composite image that enabled neuron identification with NeuroPAL, we excited mTagBFP2 using a 405 nm laser at 18% intensity under a 447/60 bandpass filter, CyOFP1 using a 488 nm laser at 32% intensity under a 585/40 bandpass filter, and mNeptune2.5 using a 637 nm laser at 50% intensity under a 655LP-TRF filter.

The rest of the image setup and experimental procedures followed the description of our previous study[67].

**Image processing.** Behavioral feature extraction and neuron registration were performed in the same way as described in our previous paper[67], with a few improvements that enabled faster and more robust extraction of calcium dynamics from dimmer infected animals:

**Improvement on the 3D segmentation U-Net:** We used a 3D U-Net for simultaneous semantic and instance segmentation. Our original U-Net[67] was trained only on the fluorescent images from SWF415 animals, which lacked the NeuroPAL construct. While it was largely able to generalize to SWF702, we were able to improve its performance further by adding four manually annotated SWF702 images to the training data. This led to an increase in the number of segmented and registered neurons in the SWF702 animals, by an average of 13 new neuron traces per animal.

**Improvement on the clustering algorithm:** In our previous work[67], we constructed a similarity matrix from the neurons detected at each time point to estimate their likelihood of being same neuron, and clustered the rows of this similarity matrix to construct the neural traces. This used to take multiple days per animal. We observed that this similarity matrix, as well as the distance for hierarchical clustering, were sparse. It would therefore be much more efficient to list out and sort all the nonzero entries of the matrix, and use a Union-Find data structure to store the clustering outputs (code available at https://github.com/flavell-lab/SparseClustering.jl). The new clustering algorithm took less than a minute per animal.

**Improvement on channel alignment of the NeuroPAL RGB composite images:** Previously[67], we performed rigid-body Euler registration using the gradient descent-based package elastix[96] to align each channel of the NeuroPAL RGB composite images. This process was susceptible to failure when the original images were initialized too far apart for the gradient to descend properly. In this current work, we implemented a custom package (code available at https://github.com/flavell-lab/euler_gpu) to perform GPU-accelerated grid search[97] across all possible combinations of rigid-body parameters to ensure that the optimal solution was always found regardless of the initial conditions. This rigid-body registration was very accurate, enabling us to skip the rest of the rigid-body and affine registrations, and use it to directly initialize the B-spline registrations, which were carried out as described previously[67].

## Neuron identification

Each registered neuron was identified as a genetically defined class using NeuroPAL manually by two independent annotators. In the case of disagreement, a third annotator would make a final decision. Neurons identified with a confidence level below 3 (on a 1–5 scale) were excluded from further analysis.

## Statistical analysis

Left/ right pair of the same neuron class was considered as the same neuron, whereas each dorsal/ ventral pair of the same neuron class was considered separately due to their distinct connection to body wall muscles. Quiescent bouts were defined as >8 consecutive seconds (or >13 consecutive fluorescent volumes) during which the animal's locomotion speed was subthreshold to measurement noise (0.01 mm/s) and no pharyngeal pumping was visible to a manual annotator. We used this bout duration threshold specifically for our brain-wide calcium imaging experiments due to the higher spatial resolution of near-infrared behavioral imaging under a 10X air objective[67] compared to a stereo microscope. The new threshold also allowed us to capture more frequent transitions between quiescence and active behavior in infected animals that were aroused from the recent mounting procedure.

For each neuron that was successfully registered across imaging time points and annotated as a genetically defined neuron class, we computed a quiescence-modulation index (QMI):

$$QMI = \frac{1}{Q}\sum_{q=0}^{Q} F_q - \frac{1}{A}\sum_{a=0}^{A} F_a$$

where $F$ refers to the GCaMP fluorescence at each imaging time point normalized by mean fluorescence of that neuron in the same recording, $Q$ refers to the total number of time points that the animal was in a quiescent bout, and $A$ refers to the total number of time points that the animal was performing active behavior. Neurons that were consistently more active during quiescence would have a positive QMI, whereas neurons that were inhibited during quiescence would have a negative QMI.

To assert statistical significance on the QMI of each neuron class, we considered the neuron classes that were observed for more than 3 times across 8 animals. We first took the mean QMI across all observations for each neuron class, and then compared this mean value

against a distribution of mean simulated QMIs computed from real neural traces of the same neuron class in relation to synthetic binary quiescence vectors. These synthetic binary quiescence vectors were generated from a hidden Markov model trained on real behavioral data of these 8 animals such that the synthetic behavioral data retained the temporal structures of the real quiescence bouts. Each neuron class then received a p-value based on the rank of its real mean QMI among 10,000 simulated mean QMIs (for instance, if the real value is either higher or lower than all simulated values, the p-value for that neuron class would be 0.0001). Since we were testing the same two-tailed hypothesis across many neuron classes simultaneously, we corrected for multiple testing using the Benjamini-Hochberg method. Neuron classes were tested significant for quiescence modulation at a false detection rate of 5%.

### Additional statistics details

Specific statistical tests used are cited in the figure legends and in some cases the Methods (where indicated). All statistical tests in the study were two-tailed unless otherwise specified. Corrections for multiple comparisons were applied when indicated in figure legends. In all figures, not significant ("n.s.") indicates a p-value > 0.05.

### Reporting summary

Further information on research design is available in the Nature Portfolio Reporting Summary linked to this article.

## Data availability

Brain-wide imaging data is available on Dryad (https://doi.org/10.5061/dryad.nvx0k6f2t). In addition, source data has been provided for this paper as an accompanying source data file. Source data are provided with this paper.

## Code availability

Brain-wide imaging related code is available on Github (https://github.com/flavell-lab/SparseClustering.jl; https://github.com/flavell-lab/euler_gpu) and has also been uploaded to Zenodo (https://doi.org/10.5281/zenodo.14991619; https://doi.org/10.5281/zenodo.14991578).

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

## Acknowledgements

We thank Mark Alkema, Yun Zhang, and members of the Flavell lab for critical reading of the manuscript. We thank the Caenorhabditis Genetics Center (supported by P40 OD010440), S. Mitani, the National BioResource Project (NBRP), P. Sternberg, D. Raizen, D. Newman, and L. Dietrich for strains; and M. Nelson and M. O'Donnell for discussion about reagents. S.P. was supported by a K. Lisa Yang Brain-Body Center Postdoctoral Fellowship. S.W.F. acknowledges funding from NIH (NS131457, GM135413); the McKnight Foundation; Alfred P. Sloan Foundation; The Picower Institute for Learning and Memory; HHMI; and The JPB Foundation. S.W.F. is an investigator of the Howard Hughes Medical Institute.

## Author contributions

Conceptualization, S.P, G.K.M, and S.W.F. Methodology, S.P and G.K.M, Software, S.P, G.K.M, A.A., A. K-L., L., J.P., and S.W.F. Formal analysis, S.P., G.K.M, and D.K. Investigation, S.P, G.K.M., D.K., E.B., U.D., T.K., M.A.G., J.D.L., and S.W.F. Writing—Original Draft, S.P, G.K.M, and S.W.F. Writing—Review & Editing, S.P, G.K.M, and S.W.F. Funding Acquisition, S.P. and S.W.F.

## Competing interests

The authors declare no competing interests.
