## [Transparent Peer Review file · Nature Communications]

Pathogen infection induces sickness behaviors through neuromodulators linked to stress and satiety in *C. elegans*

Corresponding Author: Dr Steven Flavell

Version 0:

Reviewer comments:

Reviewer #1

(Remarks to the Author)

Pradan, Madan and colleagues study the behavioral manifestations of bacterial infection in the nematode *C. elegans*. They study the mechanism of two (possibly related) behavioral phenomena. The first is a slight (about 10%) reduction in pharyngeal pumping rates in animals infected with a pathogenic variant of *Pseudomonas aeruginosa* (PA14). The second is quiescence (complete absence of body movement or feeding movements), which occurs prior to demise of the animals. They implicate the ALA neuron and three neuropeptides in the first behavior and other neurons (both pharyngeal and non-pharyngeal) as well as the FLP-13 neuropeptides in the second. The role they identified for FLP-13 peptides as suppressing quiescence and promoting survival during chronic bacterial infection is opposite of the one previously identified for these peptides in the response to acute non-infectious illness/injury. The authors followed up mutant-based analysis with interrogation of neural circuits. This line of inquiry led to the surprising conclusion that during chronic infection, there is a recruitment of various circuit and genetic systems which includes ones previously linked to stress or satiety as well as novel ones (e.g. pharyngeal neuron I5). These observations would be of general interest to an audience of neuroscientists studying neuromodulation in general and mechanisms of sickness behavior specifically. This article fits the scope for prior articles published in Nature Communications.

The authors make powerful use of tools available in this system ranging from sophisticated genetics to whole brain calcium imaging. Overall, the experiments are rigorous and the writing is clear. There are however two major issues that need clarification and a limited sets of experiments to address. I have several minor concerns, nearly all of which can be addressed by careful editing.

Major issue 1: accuracy and interpretation of pumping rate decrease

1. You show a reduction in mean pumping rate from about 320 on OP50 to about 280 on PA14. (1) how confident are you of the rates counted? this is important because the rates you show for N2 on OP50 are about 40 pumps/min faster than those reported in the literature (see table 1 in Raizen et al WormBook PMID: 23255345; and table 1 in Raizen et al, Genetics 1995). Did you (1) count pumps blinded to condition? (2) confirm that the rates you measured in real time with a manual tally counter are accurate when compared to a gold-standard measure such as by video recording analyzed in slow motion? If you have not already done this, I would ask that you count pumps with your counter while simultaneously video-recording the behavior, and then measure pumping rate off-line in slow motion. You should report the agreement between the hand tally method and the gold standard. I would also recommend you verify 2-3 of the key results using slow motion video analysis.

2. There are two ways to reduce pumping rates. One is to have fewer initiations of pumps with the same pump duration, which results in an increased inter-pump interval. The other is to reduce pumping rate by prolongation of each pump without affecting the inter-pump interval (for an example of this latter mechanism see table 1 in Lee et al J. Neuroscience 1999 . PMID: 9870947; also see Avery, JEB 1993). Distinguishing between these possibilities is important since they would lead to consideration of drastically different mechanisms.

3. Finally, it would add to the confidence of this apparent difference in pumping rate if you could compare feeding by a different method, perhaps by measuring food intake using GFP bacteria or BODIPY dye. An absence of difference in food intake would probably not be informative as these methods may not have the sensitivity to detect a 10% difference, but a positive result (i.e. reduced food intake on PA14) would be very helpful in shoring up this observation, which is foundational to the rest of the paper.

Major issue 2: assessing whether complete quiescence is a dying behavioral phenotype.

As you allude to in the final sentence of the Discussion, a major question is regarding the interpretation of the quiescence you observed before death, especially in flp-13 and dmsr-1 mutants. Is this the phenotype of a dying animal or is it distinct from the death program? Note that you observe quiescence also in immune defective mutants such as pmk-1 and tir-1.

You tried to un-couple quiescence from death by studying egl-4 lof mutants, which have reduced body movement quiescence. However in figure S4E most of the dmsr-1; egl-4 double mutant animals are already dead by 48 hours post infection, leaving very few to be observed for quiescence/not quiescence. One expt that could help you distinguish between a dying phenotype of quiescence is to follow worms longitudinally. If pausing is part of a dying program, then quiescent animals at day 1 of infection maybe more likely to die by day 2. Similarly, animals that move slower on day 1 of infection, will be more likely to be dead the next day.

I should note that even if your experiments show the quiescence to be a dying phenotype, it does not mean that it is less interesting. In fact, I would argue that behavior of a dying animal is poorly studied (with exception perhaps of the Gems lab) yet is widely relevant to all animals including to septic humans, who become stuporous before proceeding to coma and, finally, to death.

Minor

1. Figure S1B: please comment on the increased egg laying observed acutely on PA14
2. Does the observation that flp-7, flp-24 and nlp-8 are each required for pumping reduction suggest that they function in series, and not in parallel as shown for heat stress by Nath et al? More generally, how do you interpret this result?
3. Throughout the paper, you refer to a subset of pharyngeal neurons as "pharyngeal neurons" and to a subset of non-pharyngeal sensory neurons as "sensory neurons". Be careful with these classifications—there are pharyngeal neurons that are sensory and not all extra-pharyngeal sensory neurons are implicated
4. Line 255: I think I understand why you focused on I5 (as a neuron with high flp-13 expression) but unclear from fig 2E why you focus on I1 but not on M4, I3, or MI.
5. With regard to your results implicating pharyngeal neuron I5 in quiescence, I would suggest you consider a model whereby dysfunction of I5, which normally inhibits the inhibitory motor neuron M3 (see Avery, J. Exp Biol 1993 PMID: 8440973) leads to hyper-activation of the M3 neurons thereby inhibiting pumps. There would be fairly simple expts to test this model—I am NOT requesting you do such expts but at least you should mention this known role of I5 and cite Avery's paper.
6. Line 351: spell out what egl-4 encodes when you first introduce it. In general, to make the article more accessible for a non- c. elegans audience, explain what each gene encodes when you first introduce it.
7. Line 351; "reduce" is better than "prevent" since there is some quiescence during lethargus in egl-4 lof mutants. Also, have you checked whether egl-4 still has pumping quiescence without movement quiescence? This gets back to the question as to how you define "quiescence"
8. Line 367: blue light sensory responsiveness assay. Unclear why you focused only on reversals as responses. How did you consider animals who woke and accelerated in response to blue light?
9. Line 368: did you only assess animals that were fully quiescent for elevated arousal threshold? Were these animals that were not pumping or moving?
10. You use the term 'lethargy' to describe the quiescence shown by pa14 infected animals. Unless you can point to clear consensus in the literature as to the definition of lethargy, I would urge caution in using this term. For example, why use "lethargy" rather than other terms where quiescence is observed such as "sedation", "quiet wakefulness", "drowsiness" "somnia", "fatigue", etc? Since quiescence appears to be a stepping stone toward death, maybe quiescent animals are in "stupor"? Your arousal threshold experiment suggests it is not sleep. I don't think you could conclude much beyond this.
11. You describe calcium imaging experiments that failed to show activation of ALA or RIS during chronic PA14 infection. Did you do the control to ascertain that you are able to see activation of these neurons acutely after heat stress?
12. In the discussion please comment on whether you think the different mechanism of sickness quiescence you identified here is explained by the type of stressor (infection) or whether related to acuity of sickness (prior stressors such as heat shock and UV light were acute).
13. Line 576: "All animals were staged as L4s the day before." Do you mean "All animals were staged as L4s the day before the experiment"?
14. Line 585 in methods. At what temperature (or range of temp) were pumping assays performed?
15. Lines 590-591: did pumping quiescence perfectly correlate with body movement quiescence? what percent of non-pumpers were also non-movers? what percent of non-movers were also non-pumpers? also, did you consider an animal to be a non-pumper only if did not pump at all? what about animals that had 1-2 pumps in the 20 seconds of observation? In general, if "quiescence" were a distinct behavioral state, you might expect to see a bimodal distribution of pumping rates, which I do not see in figure 1.
16. Line 617: give reference for defecation counting.
17. Line 624: give more detail on how serotonin plates were made. Did you add from a concentrated stock (if yes, then in which solvent?) or did you add to molten agar?
18. Line 625: do you mean "...the day before the experiment..." ?
19. Line 693: did turning on the blue light involved a mechanical switch? If so, did you ascertain that the animals were not responding to the mechanical cue caused by flipping this switch?
20. Line 704: give more details on the custom cut PDMS corral.
21. Figure 1 legend title. Is this the best title for figure? panels I J and K are about flp-13 not ala.

22. All the figures: define what you mean by "not significant"
23. Panel 1H: I'm assuming all done 20 hours after infection. if so, pls indicate either in the fig or fig legend.
24. Panel 2E: unclear why you discuss only I1 and I5. I5 makes sense since it has high flp-13 expression but why discuss I1 and not MI, M4, and I3?
25. Panel 2F: is the I5 pharyngeal neuron acting in parallel to ASH/OLL? can you drive Cre in all three in floxed flp-13 strain to see if phenotype is enhanced?
26. Panel 2H: for heat shock cartoon, i would also include RIS (see Konietzka et al 2020 and Perez-Sanchez et al. 2021)
27. Panels 1H and 1I: seems you statistically compared the change in pumping rate in the mutant to the change in pumping rate (between OP50 and PA14) in N2. This seems awkward. Why did you not compare OP50 to PA14 pumping within genotype? I don't think this would lead to different conclusions but would improve readability of the figure.
28. How is panel 3J different from S4E? is 3J at 24 hours post-infection?
29. Panel 3J: was pumping quiescence also suppressed by egl-4 or only movement quiescence?
30. Do flp-13 mutants have reduced longevity on op50?
31. Panel S4E: death rate is much faster in dmsr-1 (and dmsr-1 double with egl-4) : at 48 hours about 80% of the worms are dead, you leave only 20% of the worm to observe quiescence. You should look at 24 hours post infection if you see quiescence (is this what you did in panel 3J?).
32. Figure 2G (and other panels): did you also express Cre under specific promoters in a strain lacking the floxed-flp13? one could imagine that expressing a high level of a DNA recombinase or over-expressing a particular promoter in the Ex arrays could impair neuron function. You may have done this control but i cannot easily find it

(Remarks on code availability)

Reviewer #2

(Remarks to the Author)

In this paper, the authors investigate the neuromodulatory systems driving sickness behaviors during infection in *C. elegans*. They utilize a model in which PA14 infection induces bouts of quiescence and a reduction in feeding in *C. elegans*. To identify the neural mechanisms underlying the PA14-induced decrease in feeding, the authors examined a panel of mutants deficient in pathways linked to stress- or pathogen-related behaviors and discovered that the gene *ceh-17* is required for the PA14-induced decrease in feeding. By studying the expression pattern of *CEH-17*, the authors further identified that the neuron ALA and its neuropeptides FLP-24, NLP-8, and FLP-7 are necessary for the infection-induced decrease in feeding. Additionally, another ALA neuropeptide, FLP-13, was found to be crucial in reducing the onset of quiescence and death.

Using sophisticated genetic manipulations, including conditional knockdowns, the authors demonstrated that FLP-13 released from sensory and pharyngeal neurons I5, I1, ASH, and OLL prevents PA14-induced quiescence and death. They also identified DMSR-1 as a receptor for FLP-13 that delays PA14-induced quiescence and death. The authors explored the immune pathways on which FLP-13 depends for the PA14-induced stress response and found that FLP-13 does not rely on any classical immune pathways for this response. They also concluded that PA14-induced quiescence does not alter the arousal threshold. Finally, the authors discovered that the neurons ASI and the neuropeptide DAF-7 are required for the PA14-induced quiescence state.

This paper represents a significant body of work and is well-organized and well-written. I only have a few comments:

-The authors screened 17 mutants and found that *ceh-17* is required for the PA14-induced decrease in feeding. Considering the vast number of stress- or pathogen-related signaling pathways, how did the authors narrow their focus to these 17 mutants?

-The phrase "sickness behaviors" in the Summary and Introduction is too general. I suggest being more specific (e.g., decreased survival, feeding, egg laying, defecation, etc).

-Unless there's a style guideline, a title for the first result section is missing.

- Figure S1A: There appears to be an increase in speed and egg laying during the first hour of infection compared to the control. Is there any link with infection?

-Is the observed decrease in feeding due to bacterial accumulation?

(Remarks on code availability)

Version 1:

Reviewer comments:

Reviewer #1

(Remarks to the Author)

The authors have nicely addressed all my concerns. Congratulations on a lovely study.

(Remarks on code availability)

Reviewer #2

(Remarks to the Author)

The authors addressed my comments

(Remarks on code availability)

We thank the reviewers and the editor for their helpful comments on our manuscript. We have revised the study to address the reviewers' comments. The main additions to the revised manuscript are: (1) new recordings and analyses to more precisely delineate the effects of PA14 infection on pharyngeal pumping, (2) new time series analyses to relate the quiescence phenotype of infected *flp-13* mutants to eventual death caused by infection, and (3) several new control experiments and descriptive experiments suggested by the reviewers. In addition, we modified the text in several places to address reviewer comments. Our point-by-point response is below. (line #s refer to the clean, revised draft)

REVIEWER COMMENTS

Reviewer #1 (Remarks to the Author):

Pradan, Madan and colleagues study the behavioral manifestations of bacterial infection in the nematode *C. elegans*. They study the mechanism of two (possibly related) behavioral phenomena. The first is a slight (about 10%) reduction in pharyngeal pumping rates in animals infected with a pathogenic variant of *Pseudomonas aeruginosa* (PA14). The second is quiescence (complete absence of body movement or feeding movements), which occurs prior to demise of the animals. They implicate the ALA neuron and three neuropeptides in the first behavior and other neurons (both pharyngeal and non-pharyngeal) as well as the FLP-13 neuropeptides in the second. The role they identified for FLP-13 peptides as suppressing quiescence and promoting survival during chronic bacterial infection is opposite of the one previously identified for these peptides in the response to acute non-infectious illness/injury. The authors followed up mutant-based analysis with interrogation of neural circuits. This line of inquiry led to the surprising conclusion that during chronic infection, there is a recruitment of various circuit and genetic systems which includes ones previously linked to stress or satiety as well as novel ones (e.g. pharyngeal neuron I5). These observations would be of general interest to an audience of neuroscientists studying neuromodulation in general and mechanisms of sickness behavior specifically. This article fits the scope for prior articles published in *Nature Communications*.

The authors make powerful use of tools available in this system ranging from sophisticated genetics to whole brain calcium imaging. Overall, the experiments are rigorous and the writing is clear. There are however two major issues that need clarification and a limited set of experiments to address. I have several minor concerns, nearly all of which can be addressed by careful editing.

Major issue 1: accuracy and interpretation of pumping rate decrease

1. You show a reduction in mean pumping rate from about 320 on OP50 to about 280 on PA14. (1) how confident are you of the rates counted? this is important because the rates you show for N2 on OP50 are about 40 pumps/min faster than those reported in the literature (see table 1 in Raizen et al *WormBook* PMID: 23255345; and table 1 in Raizen et al, *Genetics* 1995). Did you (1) count pumps blinded to condition? (2) confirm that the rates you measured in real time with a

manual tally counter are accurate when compared to a gold-standard measure such as by video recording analyzed in slow motion? If you have not already done this, I would ask that you count pumps with your counter while simultaneously video-recording the behavior, and then measure pumping rate off-line in slow motion. You should report the agreement between the hand tally method and the gold standard. I would also recommend you verify 2-3 of the key results using slow motion video analysis.

We appreciate the reviewer's suggestions and performed the suggested experiments.

First, we compared manually scored results to slowed down video quantification. For this, the experimenter manually counted pumping for 20 seconds using a tally counter. Following the manual counts, the same plate and animals were video recorded under identical conditions (free movement on agar plates with indicated bacteria). Recordings were captured at 60 frames per second and 4K resolution. (Due to technical limitations, it was not possible to simultaneously observe through the microscope eyepiece and record the videos.) This approach allowed for a close comparison between the manual counting method and the video recordings, under the environmental conditions used in the rest of the manuscript. We slowed down the video to ¼ speed and counted the pumping rate using Datavyu software. Across multiple experimental conditions, there was close agreement between the two methods for pumping quantification. None of the manual counts were lower than the video counts: 5 out of 6 conditions showed no difference at all between pumping rates quantified using the two methods (Fig S1H); in the remaining condition, the manual counts were slightly higher than the video counts (as a reminder, they were not simultaneously measured). This confirms that the pumping rates under the experimental conditions in our study are slightly higher than those in some previous studies (Raizen, Wormbook and Raizen, Gen, 1995), as noted by the reviewer. Our recording conditions involved low-peptone plates and specific densities of food, as described in the Methods. Indeed, pumping rates can vary significantly with bacterial densities as reported by Suk Lee, et al (Nat Comm, 2017), which could contribute to these differences.

Second, as suggested by the reviewer, we took this opportunity to repeat several key experiments from our study using the gold standard slowed-down-video pumping quantification. Specifically, we tested three genotypes (WT, *ceh-17*, *flp-13*) on both OP50 and PA14. All of these results matched our prior work, confirming the effect of PA14 on WT pumping and the *ceh-17* and *flp-13* mutant phenotypes.

Third, we have added a description to the methods section that the vast majority of the pumping data quantified in the study was scored blind to genotype. In addition, all key experiments were repeated by two independent scorers (S.P and G.M.). A small number of early pilot experiments as we began the study were not scored blinded, but we felt that this was not a sufficient criterion for data exclusion, so all scored data is included in the manuscript.

We have added these results in Supp Fig 1H. In addition, we provide Supp Video Files 1-2 as examples (*flp-13* on OP50 and PA14, respectively). Discussion of these new experiments is in lines 156-163. Description of experiment blinding is in lines 639-641.

2. There are two way to reduce pumping rates. One is to have fewer initiations of pumps with the same pump duration, which results in an increased inter-pump interval. The other is to reduce pumping rate by prolongation of each pump without affecting the inter-pump interval (for an example of this latter mechanism see table 1 in Lee et al J. Neuroscience 1999 . PMID: 9870947; also see Avery, JEB 1993). Distinguishing between these possibilities is important since they would lead to consideration of drastically different mechanisms.

We performed this suggested analysis. Using the slowed-down-video pumping quantification method, we got timestamps for every grinder contraction for WT animals on OP50 and PA14. We measured the time intervals between consecutive grinder contractions and found that there was a shift in the distribution of inter-pump-intervals, such that PA14-infected animals had longer inter-pump-intervals (Fig S1I). We next examined whether these longer intervals were due to prolonged pharyngeal contraction vs normal duration contractions followed by prolonged relaxation. To do so, we quantified pharyngeal contractions at high temporal resolution – in videos played back at 1/8 speed (we note that this is not as precise as electropharyngeograms, but we wanted to be sure to record this under identical environmental conditions to those used in the other experiments in our study). While the time intervals between grinder contractions and relaxations were short, the pauses after relaxation (before the next pump) were clearly longer than on standard OP50, often lasting hundreds of milliseconds. This suggests that PA14 infection leads to fewer pump initiations. These data are shown in Fig. S1J.

We thank the reviewer for pointing out this interesting possibility of differences in underlying mechanisms. We do wish to note that it is technically challenging to acquire high resolution videos in freely moving animals on dense bacterial lawns to accurately measure all movements in the pharynx. Therefore, we indicate some caution in interpretation of these data in the manuscript and suggest that more precise analyses (electropharyngeogram, etc.) may be necessary to fully unpack the precise effects of PA14 on all pharyngeal movements. This is in lines 160-163.

3. Finally, it would add to the confidence of this apparent difference in pumping rate if you could compare feeding by a different method, perhaps by measuring food intake using GFP bacteria or BODIPY dye. An absence of difference in food intake would probably not be informative as these methods may not have the sensitivity to detect a 10% difference, but a positive result (i.e. reduced food intake on PA14) would be very helpful in shoring up this observation, which is foundational to the rest of the paper.

We performed this suggested experiment using OP50-GFP bacteria. Specifically, we compared ingestion of OP50-GFP in control animals vs PA14-infected animals. The overall fluorescence reported by ingested bacteria was very low across the gut and pharynx of these animals and this method did not show a difference between the two conditions. As the reviewer mentioned, an absence of difference, with such low counts of fluorescence, is hard to interpret as it could be a false negative due to the sensitivity of the assay. Nevertheless, we have added these results to the manuscript with the proper description and interpretation. We have added these results

in Fig S1K and discussed them in lines 669-673.

Major issue 2: assessing whether complete quiescence is a dying behavioral phenotype.

As you allude to in the final sentence of the Discussion, a major question is regarding the interpretation of the quiescence you observed before death, especially in *flp-13* and *dmsr-1* mutants. Is this the phenotype of a dying animal or is it distinct from the death program? Note that you observe quiescence also in immune defective mutants such as *pmk-1* and *tir-1*.

You tried to un-couple quiescence from death by studying *egl-4* lof mutants, which have reduced body movement quiescence. However in figure S4E most of the *dmsr-1*; *egl-4* double mutant animals are already dead by 48 hours post infection, leaving very few to be observed for quiescence/not quiescence. One expt that could help you distinguish between a dying phenotype of quiescence is to follow worms longitudinally. If pausing is part of a dying program, then quiescent animals at day 1 of infection maybe more likely to die by day 2. Similarly, animals that move slower on day 1 of infection, will be more likely to be dead the next day.

I should note that even if your experiments show the quiescence to be a dying phenotype, it does not mean that it is less interesting. in fact, i would argue that behavior of a dying animal is poorly studied (with exception perhaps of he Gems lab) yet is widely relevant to all animals including to septic humans, who become stuporous before proceeding to coma and, finally, to death.

We agree that this is an interesting issue, and so we carried out several new experiments. We tested the *dmsr-1*;*egl-4* mutants at both 20 hours and 48 hours. We report the 20 hour phenotype in Fig 3J, when 80-90% of the animals are still alive. We did not observe any quiescence in these *egl-4* or *dmsr-1*;*egl-4* double mutants while *dmsr-1* mutants already had a substantial fraction of quiescent animals. We kept these animals for an additional day on PA14 and observed the same animals again at 48 hours (as reported in original Fig S4E, currently Fig S5E). Even at this later time point, when most *dmsr-1* animals are dead, we did not observe any quiescence in *dmsr-1*;*egl-4* animals. Anecdotally, at late stages of infections, we observed very sick, almost decaying *dmsr-1*;*egl-4* animals that were still pumping, even though wild-type animals are almost never pumping at this advanced stage of infection. These results suggest that quiescence is often followed by death, but is still separable: animals can die without exhibiting observable quiescent states (i.e. in the *egl-4* mutant background).

We also performed the longitudinal analysis suggested by the reviewer. Here, we followed 34 individual *flp-13* animals from first exposure to PA14 until their death, assaying their behavior at specific timepoints along the way (Fig S6). This longitudinal study revealed that approximately half of the animals showed transitions from normal behavior (pumping and moving) to quiescence (no pumps or movement observed in 20 secs) and then reversed back to normal behavior. A similar fraction of animals transitioned from normal to quiescent states and died without any observed transition back to normal behavior. A few animals died without showing observable quiescence (though we note that they could have displayed quiescence between assay timepoints). Overall, this analysis further suggests that while quiescence is

generally observed in sick animals that will eventually die, it is not a strict predecessor of death. We describe these results in lines 401-411 and they are shown in Fig. S6. These results are also consistent with our observation that this quiescence can be acutely reversed through optogenetic stimulation of the arousing PDF peptidergic pathway (Fig 3H-I).

Minor

1. Figure S1B: please comment on the increased egg laying observed acutely on PA14

As Reviewer 2 also expressed interest in this initial increase in egg-laying, we tested whether the increase in egg-laying in the first hour of infection (Fig S1B) requires infectious bacteria. Specifically, we tested egg-laying during the first hour on UV-killed PA14, and did not see the increase observed in infectious PA14. This suggests that the increase in egg-laying in the first hour of infection requires infectious bacteria. We have added these new results in Fig S1E and, as suggested, expanded our discussion of these results in lines 116-120.

2. Does the observation that *flp-7*, *flp-24* and *nlp-8* are each required for pumping reduction suggest that they function in series, and not in parallel as shown for heat stress by Nath et al? More generally, how do you interpret this result?

We find that the single mutants of these genes do not show a reduction in pumping rates upon infection and agree with the reviewer's interpretation that they seem to act in series. While this is different from parallel pathway identified by Nath et al., functions of these peptides may be different in the heat shock versus infection contexts. We find that functionally critical release sites of these peptides can differ depending on if animals have been heat-shocked or infected (Fig 2). We now further discuss this in lines 503-515 of the manuscript.

3. Throughout the paper, you refer to a subset of pharyngeal neurons as "pharyngeal neurons" and to a subset of non-pharyngeal sensory neurons as "sensory neurons". Be careful with these classifications—there are pharyngeal neurons that are sensory and not all extra-pharyngeal sensory neurons are implicated.

We have clarified this language and now use the term "non-pharyngeal sensory neurons" to refer to these cells.

4. Line 255: I think I understand why you focused on I5 (as a neuron with high *flp-13* expression) but unclear from fig 2E why you focus on I1 but not on M4, I3, or MI.

We apologize that we were not clearer about this in the original manuscript. We used cell-specific Cre lines to disrupt *flp-13* expression in subsets of neurons. Expression of *flp-13* was disrupted in M4 in a strain with Cre driven by the *ceh-28* promoter. This strain did not show significantly different quiescence compared to the Cre- control. However, when Cre was driven by the *ceh-53* promoter, disrupting *flp-13* expression from both I5 and M4, we observed increased quiescence. We hence concluded that I5 was a relevant source for *flp-13* for inducing quiescence. Similarly, by comparing the effects of the *ceh-45* and *degt-1* promoters, we infer

that I1 is a functionally relevant source of flp-13.

5. With regard to your results implicating pharyngeal neuron I5 in quiescence, I would suggest you consider a model whereby dysfunction of I5, which normally inhibits the inhibitory motor neuron M3 (see Avery, J. Exp Biol 1993 PMID: 8440973) leads to hyper-activation of the M3 neurons thereby inhibiting pumps. There would be fairly simple expts to test this model—I am NOT requesting you do such expts but at least you should mention this known role of I5 and cite Avery's paper.

We have added a reference to this work and discussion of this idea in lines 535-536.

6. Line 351: spell out what egl-4 encodes when you first introduce it. In general, to make the article more accessible for a non- *C. elegans* audience, explain what each gene encodes when you first introduce it.

We have added this in line 378, and have gone through the manuscript to make the *C. elegans* jargon more broadly understandable.

7. Line 351; "reduce" is better than "prevent" since there is some quiescence during lethargus in egl-4 lof mutants. Also, have you checked whether egl-4 still has pumping quiescence without movement quiescence? This gets back to the question as to how you define "quiescence"

We have changed the text to "reduce" in line 378. egl-4 animals did not show any quiescence, movement or pumping, in the infection context. In the manuscript, we call animals quiescent only when both feeding and locomotion are absent in the 20 sec observation window. However, during scoring we recorded notes about whether animals exhibited only pumping or only movement. Looking back at these notes, this did not occur in egl-4 animals.

8. Line 367: blue light sensory responsiveness assay. Unclear why you focused only on reversals as responses. How did you consider animals who woke and accelerated in response to blue light?

We titrated the light intensity to a low level that reliably induced reversals in uninfected WT control animals, but did not elicit strong acceleration. We then used this low light intensity to test all other conditions. Due to assay conditions being set up in the way, we only quantified the reliable reversal response. We now note this in the Methods in lines 794-797.

9. Line 368: did you only assess animals that were fully quiescent for elevated arousal threshold? Were these animals that were not pumping or moving?

Yes, we assessed animals that were completely quiescent only, meaning that they showed both an absence of locomotion and were not pumping. We have added this detail to the Methods section describing the assay (lines 642-651).

10. You use the term ‘lethargy’ to describe the quiescence shown by pa14 infected animals. Unless you can point to clear consensus in the literature as to the definition of lethargy, I would urge caution in using this term. For example, why use “lethargy” rather than other terms where quiescence is observed such as “sedation”, “quiet wakefulness”, “drowsiness” “somnolence”, “fatigue”, etc? Since quiescence appears to be a stepping stone toward death, maybe quiescent animals are in “stupor”? Your arousal threshold experiment suggests it is not sleep. I don’t think you could conclude much beyond this.

We chose to use the term “lethargy” to differentiate this from “sleep”. We agree with the reviewer that given the arousal threshold experiment, we could not call this “sleep”. However, our data are not consistent with this form of quiescence being a strict predecessor of death either: it is acutely reversible (Fig 3H and I) and animals can reverse back to normal movement and feeding even when undisturbed (Fig S6). Lethargy seemed like a cautious word choice (though we are open to changing it). Either way, we thought it made sense to discuss this word choice in the revised Discussion and suggest that further studies on this quiescence state could prompt a revision of the terminology in the future (lines 556-559).

11. You describe calcium imaging experiments that failed to show activation of ALA or RIS during chronic PA14 infection. Did you do the control to ascertain that you are able to see activation of these neurons acutely after heat stress?

We have not run controls to see activation of ALA or RIS upon heat stress. This would be extremely challenging on our current whole-brain imaging platform. While validating these prior results from others in the field could be valuable, we felt that it was beyond the scope of our study.

12. In the discussion please comment on whether you think the different mechanism of sickness quiescence you identified here is explained by the type of stressor (infection) or whether related to acuity of sickness (prior stressors such as heat shock and UV light were acute).

We have commented on this in lines 563-569.

13. Line 576: “All animals were staged as L4s the day before.” Do you mean “All animals were staged as L4s the day before the experiment”?

Yes, we have clarified this language in line 626.

14. Line 585 in methods. At what temperature (or range of temp) were pumping assays performed?

All experiments were performed at room temperature, between 22-24C.

15. Lines 590-591: did pumping quiescence perfectly correlate with body movement quiescence? what percent of non-pumpers were also non-movers? what percent of non-movers were also

non-pumpers? also, did you consider an animal to be a non-pumper only if did not pump at all? what about animals that had 1-2 pumps in the 20 seconds of observation? In general, if “quiescence” were a distinct behavioral state, you might expect to see a bimodal distribution of pumping rates, which I do not see in figure 1.

We call an animal quiescent only when both feeding and locomotion are entirely absent during the 20 secs of observation. If animals had 1-2 pumps, they were not categorized as quiescent. For *flp-13* mutants (whose quiescence we have examined the most times), the majority of non-pumpers were also non-movers and vice versa. Quantifying a representative experiment, the breakdown of phenotypes in infected *flp-13* mutants was: 31% yes-moving/yes-feeding; 42% no-moving/no-feeding; 2% yes-moving/no-feeding; 16% no-moving/yes-feeding; 9% dead. We have added this information to our description of the assay in lines 646-650.

16. Line 617: give reference for defecation counting.

We have added this reference in line 699.

17. Lin 624: give more detail on how serotonin plates were made. Did you add from a concentrated stock (if yes, then in which solvent?) or did you add to molten agar?

We made a 50mM stock solution of serotonin creatinine sulfate monohydrate powder (Sigma - H7752) in water and added it to NGM media in the molten stage right before pouring plates. We now describe this in the methods section, lines 721-722.

18. Line 625: do you mean “...the day before the experiment...” ?

Yes, we have clarified this language in line 723.

19. Line 693: did turning on the blue light involved a mechanical switch? If so, did you ascertain that the animals were not responding to the mechanical cue caused by flipping this switch?

We thank the reviewer for suggesting this important control. This was an unlikely possibility since the switch for the light is located on the barrel of the microscope, close to the eyepiece but far away from the stage. Nevertheless, we tested this for 16 quiescent animals over three different days and found that none of the animals were roused by the simple mechanical disturbance of flipping this switch (with the light source off). We have added a description of this to the Methods section where we describe the assay (lines 803-807).

20. Line 704: give more details on the custom cut PDMS corral.

We have added additional details in the methods section, lines 813-815.

21. Figure 1 legend title. Is this the best title for figure? panels I J and K are about *flp-13* not *ala*.

We have changed the title for figure 1 to better describe the figure.

22. All the figures: define what you mean by “not significant”

All conditions marked as “not significant” indicate that p-values were greater than 0.05 for the statistical tests run for that specific analysis. This is now indicated in an Additional Statistical Details section of the Methods (lines 928-931).

23. Panel 1H: I’m assuming all done 20 hours after infection. if so, pls indicate either in the fig or fig legend.

Yes, these were done at 20 hours post infection. We have clarified this in the figure legend (Fig. 1H).

24. Panel 2E: unclear why you discuss only I1 and I5. I5 makes sense since it has high flp-13 expression but why discuss I1 and not MI, M4, and I3?

We have addressed this in response to point 4 in the reviewer’s comments (above).

25. Panel 2F: is the I5 pharyngeal neuron acting in parallel to ASH/OLL? can you drive Cre in all three in floxed flp-13 strain to see if phenotype is enhanced?

We attempted to identify neurons which have critical roles in inducing quiescence and did not run a full set of experiments to test the full set of relationships (series/parallel) between these neurons. We haven’t tested all the possible combinations of Cre lines that would be required to fully establish such relationships.

26. Panel 2H: for heat shock cartoon, i would also include RIS (see Konietzna et al 2020 and Perez-Sanchez et al. 2021)

We have added the RIS neuron to this cartoon in Fig 2H and Fig 5 and added citations for Konietzna et al, 2020 and Chavez-Perez et al, 2021.

27. Panels 1H and 1I: seems you statistically compared the change in pumping rate in the mutant to the change in pumping rate (between OP50 and PA14) in N2. This seems awkward. Why did you not compare OP50 to PA14 pumping within genotype? I don’t think this would lead to different conclusions but would improve readability of the figure.

The reviewer’s intuition is indeed correct here. We ran statistics (non-parametric Wilcoxon test) to test differences between OP50 and PA14 conditions for all strains at 20 hours for Fig 1H and 1I and that gives us the same results as we report. However, we believe that it’s more appropriate to test whether the change in pumping in the mutant is significantly smaller than the change in pumping in wild-type. This article convinced us to use that approach: Nieuwenhuis et

al. Nat Neurosci. 2011 Aug 26;14(9):1105-7. doi: 10.1038/nn.2886. PMID: 21878926.

28. How is panel 3J different from S4E? is 3J at 24 hours post-infection?

Panel 3J is a quantification of the same animals as Fig S5E (formerly, S4E), where 3J indicates fraction of quiescent animals at 20 hours, and Fig S5E indicates the same at 48 hours. We have modified the text (lines 378-392) and figure legend to make this more apparent, as we recognize that it was hard to appreciate the difference in the original manuscript.

29. Panel 3J: was pumping quiescence also suppressed by *egl-4* or only movement quiescence?

Both pumping quiescence and movement quiescence were suppressed in *egl-4* mutants.

30. Do *flp-13* mutants have reduced longevity on *op50*?

We ran new experiments to test this and found that across three independent replicates *flp-13* mutants surprisingly have increased longevity on OP50 compared to wild-type animals (Fig. S3B). Similar results have also been reported recently in Rathor, *et al.*, 2024, where RNAi suppression of *flp-13* expression led to increased longevity (this other study did not examine *flp-13* phenotypes on PA14). These results are discussed in lines 248-253.

31. Panel S4E: death rate is much faster in *dmsr-1* (and *dmsr-1* double with *egl-4*) : at 48 hours about 80% of the worms are dead, you leave only 20% of the worm to observe quiescence. You should look at 24 hours post infection if you see quiescence (is this what you did in panel 3J?).

We provide this analysis in Fig 3J, which shows that even at earlier timepoints with mostly live animals *egl-4* suppresses quiescence.

32. Figure 2G (and other panels): did you also express Cre under specific promoters in a strain lacking the floxed-*flp13*? one could imagine that expressing a high level of a DNA recombinase or over-expressing a particular promoter in the Ex arrays could impair neuron function. You may have done this control but i cannot easily find it

We ran this control experiment suggested by the reviewer. Specifically, we injected Cre pan-neuronally in a wild-type background and found that this strain behaves similarly to wild-type animals upon infection (please refer to lines 263-265 and Fig S3F).

Reviewer #2 (Remarks to the Author):

In this paper, the authors investigate the neuromodulatory systems driving sickness behaviors during infection in *C. elegans*. They utilize a model in which PA14 infection induces bouts of quiescence and a reduction in feeding in *C. elegans*. To identify the neural mechanisms underlying the PA14-induced decrease in feeding, the authors examined a panel of mutants

deficient in pathways linked to stress- or pathogen-related behaviors and discovered that the gene *ceh-17* is required for the PA14-induced decrease in feeding. By studying the expression pattern of CEH-17, the authors further identified that the neuron ALA and its neuropeptides FLP-24, NLP-8, and FLP-7 are necessary for the infection-induced decrease in feeding. Additionally, another ALA neuropeptide, FLP-13, was found to be crucial in reducing the onset of quiescence and death.

Using sophisticated genetic manipulations, including conditional knockdowns, the authors demonstrated that FLP-13 released from sensory and pharyngeal neurons I5, I1, ASH, and OLL prevents PA14-induced quiescence and death. They also identified DMSR-1 as a receptor for FLP-13 that delays PA14-induced quiescence and death. The authors explored the immune pathways on which FLP-13 depends for the PA14-induced stress response and found that FLP-13 does not rely on any classical immune pathways for this response. They also concluded that PA14-induced quiescence does not alter the arousal threshold. Finally, the authors discovered that the neurons ASI and the neuropeptide DAF-7 are required for the PA14-induced quiescence state.

This paper represents a significant body of work and is well-organized and well-written. I only have a few comments:

-The authors screened 17 mutants and found that *ceh-17* is required for the PA14-induced decrease in feeding. Considering the vast number of stress- or pathogen-related signaling pathways, how did the authors narrow their focus to these 17 mutants?

We surveyed the literature for genes linked to PA14 infection and regulation of feeding, which resulted in this list of 17 mutants. This list is by no means exhaustive. Once we found and validated an interesting phenotype (in *ceh-17* mutants), we did not screen further mutants. We describe the rationale for selecting these 17 mutants more clearly now in lines 170-175 of the revised manuscript.

-The phrase “sickness behaviors” in the Summary and Introduction is too general. I suggest being more specific (e.g., decreased survival, feeding, egg laying, defecation, etc).

We agree that this can be vague and so we dramatically cut our usage of the term “sickness behaviors” in the revised manuscript (used 3 times now, compared to 14 times in the last draft). We now only use the term “sickness behaviors” when describing how multiple behaviors change in concert during an infection, as it is commonly used in the mammalian field. We agree that in almost all cases, it was preferred to simply use more specific terms (feeding reduction, etc) depending on the assay at hand.

-Unless there's a style guideline, a title for the first result section is missing.

We appreciate the reviewer noticing this. We have now added a title for the first section (line 97).

- Figure S1A: There appears to be an increase in speed and egg laying during the first hour of infection compared to the control. Is there any link with infection?

There is no increase in speed during the first hour of infection (Fig S1A), but there is indeed a significant increase in egg-laying. Based on the reviewer's question, we tested whether the increase in egg-laying in the first hour of infection (Fig S1B) requires infectious bacteria. Specifically, we tested egg-laying during the first hour on UV-killed PA14 and did not see the increase observed in infectious PA14 (Fig S1E). This suggests that the increase in egg-laying in the first hour of infection is a response to infectious bacteria. **We have added these results in Fig S1E and referenced them in lines 116-120.**

-Is the observed decrease in feeding due to bacterial accumulation?

This is a very interesting question, as it is possible that increased bacterial accumulation in the gut during PA14 infection might trigger a feeding reduction. In new experiments, we examined three different mutants that exhibit increased accumulation of non-infectious bacteria due to deficits in expulsion/defecation (Das, eLife, 2024). We tested whether these animals exhibit reduced feeding on OP50 bacteria, as a general test of whether increased bacterial accumulation leads to reduced feeding. Indeed, we found that all three of the mutants have decreased feeding compared to wild-type animals on OP50. This raises the possibility that increased accumulation of bacteria can indeed play a role in decreasing feeding behavior. However, we note that this does not definitively show that that is the sole mechanism by which PA14 infection leads to a decrease in feeding. Nevertheless, this new addition to the study provides an interesting lead for an underlying mechanism moving forwards. **We have added these results in Fig S1L and discuss them in lines 163-169.**